# Expected Annual Minima from an Idealized Moving Average Drought Index

James H.Stagge[1], Kyungmin Sung[1,2], Irenee Munyejuru[1]; Md Atif Ibne Haidar[1]

[1] Department of Civil, Environmental and Geodetic Engineering, The Ohio State University, Columbus, OH, 43210, USA

[2] Korea Adaptation Center for Climate Change, Korea Environment Institute, Sejong, Republic of Korea

*Correspondence to*: James H. Stagge (stagge.11@osu.edu)

**Abstract.** Numerous drought indices originate from the Standardized Precipitation Index (SPI) and use a moving average structure to quantify drought severity by measuring normalized anomalies in hydroclimate variables. This study examines the theoretical probability of annual minima from such a process. To accomplish this, we derive a stochastic model and use it to simulate 10 million years of daily or monthly SPI values in order to determine the distribution of annual exceedance probabilities. We believe this is the first explicit quantification of annual extreme exceedances from a moving average
process where the moving average window is proportionally large (5-200%) relative to the year, as is the case for many moving window drought indices. The resulting distribution of annual minima follow a Generalized Normal distribution, rather than the Generalized Extreme Value (GEV) distribution, as would be expected from extreme value theory. From a more applied perspective, this study provides the expected annual return periods for the SPI or related drought indices with common accumulation periods (moving window length), ranging from 1 to 24 months. We show that the annual return
period differs depending on both the accumulation period and the temporal resolution (daily or monthly). The likelihood of exceeding an SPI threshold in a given year decreases as the accumulation period increases. This study provides clarification and a caution for the use of annual return period terminology (e.g. the 100 year drought) with the SPI and a further caution for comparing annual exceedances across indices with different accumulation periods or resolutions. The study also distinguishes between theoretical values, as calculated here, and real-world exceedance probabilities, where there may be
climatological autocorrelation beyond that created by the moving average.

# 1 Introduction

The Standardized Precipitation Index (SPI) (Guttman, 1999; McKee et al., 1993) is used to measure meteorological drought operationally by many organizations, including the WMO and numerous drought monitors (Cammalleri et al., 2021; Hao et

al., 2014; Heim and Brewer, 2012; Lawrimore et al., 2002; Sheffield et al., 2014; Svoboda et al., 2002; World Meteorological Organization (WMO) and Global Water Partnership (GWP), 2016). This index is particularly useful because it requires only precipitation data and mirrors commonly agreed-upon definitions of meteorological drought; a sustained and spatially extensive period of below-average water availability (Heim, 2002; Lloyd-Hughes, 2014; Tallaksen and Van Lanen, 2004). The SPI measures accumulated or mean precipitation during a moving window and normalizes this quantity relative

to the historical climatology for that day of the year, thereby producing a normalized anomaly. The SPI is typically referred to using its accumulation period, the backwards looking moving window, measured in months. The SPI-3 therefore represents precipitation anomalies based on the previous 3 months. The SPI can also be calculated at a daily temporal resolution, though the naming convention still typically refers to months, for example using a 90-day moving window to calculate the SPI-3. Accumulated precipitation is bounded by zero and typically positively skewed, commonly leading to the

use of independently calibrated gamma distributions (Guttman, 1999; Lloyd-Hughes and Saunders, 2002; Stagge et al., 2015; Stagge and Sung, 2022) to represent climatology and transform accumulated precipitation into percentiles. These percentiles are ultimately transformed to anomalies of the standard normal distribution, with mean of 0 and standard deviation of 1.

The fundamental SPI concept has since been expanded to a family of normalized drought indices, each quantifying anomalies from different portions of the hydrologic cycle. For example, normalized drought indices have measured anomalies in the climatic water balance, soil moisture, groundwater, and streamflow, referred to respectively as the: Standardized Precipitation Evapotranspiration Index (SPEI, Beguería et al., 2013, Vicente-Serrano et al., 2010), the Standardized Soil Moisture Index (SSI, Sheffield et al., 2004), Standardized Groundwater Level Index (SGI, Bloomfield & 

Marchant, 2013) and the Standardized Runoff Index (SRI, Shukla & Wood, 2008). The principles developed in this study are applicable to all normalized drought indices that employ a moving average structure, but we will occasionally refer to the SPI as the simplest example of this broader class. It should be noted that these indices sometimes use instantaneous values, for which a moving window is not applied and the findings in this study are less relevant.

Values from normalized drought indices follow the standard normal distribution (mean = 0, standard deviation = 1) within the reference calibration period, resulting in relatively interpretable percentiles. These percentiles have then been used to develop thresholds. For example, the US Drought Monitor uses five categories classified from D0 to D4 based on SPI thresholds of -0.5, -0.8, -1.3, -1.5, and -2 (Xia et al., 2014). These thresholds correspond very roughly with percentiles of $\leq$ 30%, 20%, 10%, 5%, and 2% which allow them to be compared with other percentile-based indices. In this study, we will

use the USDM SPI thresholds as reference points, though using the exact corresponding percentiles. For example, an extreme drought (D3) is assumed to occur for SPI values of -1.5 to -2, or 1.5 to 2 standard deviations drier than typical for that time of year. The likelihood of a given day falling within the D3 category is therefore 4.4%, the difference between exceedance probabilities for -1.5 (6.7%) and -2 (2.3%). These thresholds have statistical utility, but are arbitrary. For example, different thresholds have been used previously (McKee et al., 1993), drought impacts are linked to a wide range of

thresholds (Blauhut et al., 2016; Stagge et al., 2014), and the US Drought Monitor's blended Objective Drought Indicator uses additional logic to classify droughts (Anderson et al., 2013; Xia et al., 2014).

Confusion of interpretation can occur with regards to normalized drought indices because hydrologists often quantify extremes in terms of annual exceedance probabilities, or their reciprocal, the return period. However, annual exceedance

probabilities differ from the probability associated with a normalized drought index series because each day follows a standard normal distribution. For example, there is a 6.7% chance that the SPI on January 1 will fall into the D3 category or worse (see above), but there is also a 6.7% chance of falling into this drought category on January 2nd, and all subsequent days of the year. Therefore, it is not correct to state that there is a 6.7% chance of experiencing a D3 drought or worse (one SPI value less than -1.5) in a given year. The latter statement is what hydrologists typically define as return period or annual

exceedance probability, which is clearly distinct from the daily or monthly SPI probabilities. If SPI values were independent and identically distributed (i.i.d.), the likelihood of a year with a single SPI value below a given threshold would be given by:

$$1 - (1 - p)^n ,$$ (1)

where p is the probability of exceedance for each time step and n is the length of the sample period: 12 months or 365 days.

The probability of at least one observation being below -1.5 in a given year then approaches 100% for daily time steps and is 56.4% for monthly time steps. However, SPI time series are fundamentally not i.i.d., instead being subject to a large amount of temporal autocorrelation due to the SPI's moving average structure. The degree of temporal autocorrelation depends on the length of the moving window or accumulation period, which can be tuned to alternatively capture short droughts with smaller periods or capture seasonal to multi-year droughts using longer periods. While temporal autocorrelation invalidates

annual exceedance estimates from Eq. 1, the moving average structure provides a predictable and pre-defined structure that can be leveraged to quantify the likelihood of annual exceedances.

Despite a robust field of research into moving average models as part of the autoregressive-moving average (ARMA) family of time series models (Box and Jenkins, 1970; Wilks, 2011), we were unable to identify prior research quantifying the

extreme behaviour of a moving average sequence where the moving window is long relative to the time interval, as is the case for normalized drought indices. Solutions exist for the simple AR(1) case (Hirtzel, 1985a, 1985b) and that of an AR(n) case, although the latter requires high dimension copulas and are unstable (Tsoukalas, 2022). Prior extreme value theory for

moving averages (Davis and Resnick, 1991; Rootzen, 1986) appears to break down under the conditions of the normalized drought indices, as we outline here. The purpose of this study is therefore to quantify the expected annual minima, and their associated return periods, from a theoretical moving average series when simulated from daily or monthly sequences with moving windows that mimic common lengths for the SPI and other drought indices. Before simulating these annual minima, we first derive a stochastic model for an idealized moving average series and then show how, for several example sites, temporal autocorrelation from observations broadly follow this idealized model due to the underlying moving average. For the remainder of this study, we will focus on annual minima, as relevant for droughts; however, all findings apply equally for positive (maxima) extremes because the standard normal distribution is symmetrical.

## 2 Methods

This study is organized by first developing a simulation method to generate long sequences that meet two criteria: (1) values for each day or month follow the standard normal distribution with mean of 0 and standard deviation of 1, and (2) values follow a uniformly weighted backwards looking moving average. Following this derivation, we simulate extremely long sequences from this process and extract the annual minima for multiple thresholds. We also estimate the first four moments of the annual minima and test several extreme value distributions to determine whether the annual minima can be adequately represented by a continuous probability distribution. Using these fitted distributions, we describe the annual exceedance probabilities for any threshold.

This approach is designed only to explore the theoretical behaviour of a simplified case; one affected only by the structural persistence caused by the moving average. This approach does not consider additional climatological persistence caused by a region's climatology or by macroscale drivers like atmospheric teleconnections. We make this distinction between structural and climatological persistence throughout the study. Normalized drought indices in the real world are impacted by a combination of these and other factors (see Discussion), requiring site- and index-specific analyses. But, as we show for a variety of case study sites across climate regions, structural persistence typically represents the vast majority of temporal autocorrelation. We therefore present here the solution to the limiting case of structural persistence only for clarification and as a benchmark for future comparisons.

### 2.1 Relating SPI to Moving Average processes

The SPI calculates a moving average or moving sum of precipitation, which is then normalized for each day or month of the year relative to its historical climatology (Guttman, 1999; McKee et al., 1993). For the purposes of this study, we stipulate that the simplest SPI time series is also a moving average series following the necessary standard normal distribution with mean of 0 and standard deviation of 1. The potential implications of this assumption are explored further in the Discussion.

A random series of SPI-q values, where q is the accumulation period, can therefore be generated using random daily or monthly incremental changes, called innovations, $Z_t$, sampled from:

$Z_t \sim N(0, \sqrt{q})$      i.i.d.                                                                        (2)

which progressively enter and leave a moving average:

$$SPI(t) = \frac{1}{q} \sum_{j=0}^{q} Z_{t-j}$$                                                                    (3)

to generate an *SPI(t)* series. This produces the requisite standard normal distribution, while maintaining a moving window of q time steps. The SPI has a backwards looking "memory" of q time steps (days or months), where each is weighted equally. Within the Autoregressive Moving Average (ARMA) framework (Box and Jenkins, 1970; Wilks, 2011), such a model can be written as a Moving Average process, MA(q-1). In this notation, the moving window is written as q-1, rather than q, because q-1 represents the number of time lags in addition to the innovation added in the current step, i.e. at a time lag of zero.

Writing the model in standard ARMA notation allows the application of a deep body of literature regarding the properties of ARMA models. Using this standard notation, the MA(q-1) process has q-1 MA coefficients of 1 and innovations of $\sqrt{q}/q$. For example, it is possible to simulate an SPI-6 sequence using an MA(5) with MA coefficients of [1,1,1,1,1] and innovations randomly sampled from an i.i.d. Gaussian distribution $N(0, \frac{\sqrt{6}}{6})$.

The autocorrelation function (ACF) for such a theoretical SPI-q series, represented by an MA(q-1) process, has a linear decay of 1/q per time lag (Fig 1). The ACF becomes zero past q lags because these innovations are no longer part of the moving average. Expanding on the previous SPI-6 example, temporal autocorrelation falls from 1 at lag 0 to 0.1666 at lag 5, followed by zero autocorrelation for lags of 6 months and beyond (Fig. 1). The same would occur for a 6 month SPI-6 series using a daily temporal resolution, but autocorrelation would decay linearly towards zero after 182 days (approximately 6

months)

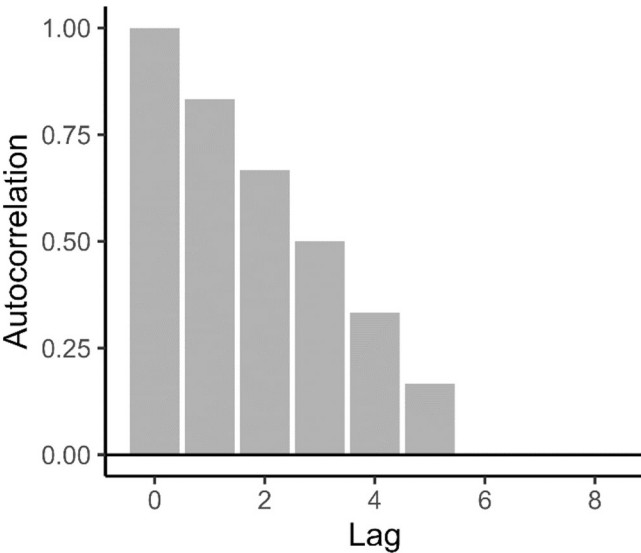

**Figure 1: Autocorrelation function (ACF) for an SPI-6 generating process.**

Moving average processes with discrete time lags, like the SPI, can be converted to an infinite-order autoregressive process if the MA model is invertible (Granger and Andersen, 1978). For an ARMA process to be invertible, all roots of the characteristic polynomial must lie outside the complex unit circle (> 1) (Davidson, 1981; Granger and Andersen, 1978; Hallin, 1984). Roots of the MA(q-1) process defined here are exactly on the unit circle, making this theoretical model not invertible. Therefore, there is no equivalent AR($\infty$) model.

**2.2 Stochastic Simulation from MA Model**

Normalized drought index time series were simulated using the moving average model (Eq. 2 and 3), with unique simulations run using daily and monthly temporal resolutions. Monthly simulations were performed using accumulation periods of 1, 2, 3, 6, 9, 12, and 24 months to address the most commonly used moving windows for the SPI. Daily simulations recreated these accumulation periods with an additional half-month window: 15, 30, 60, 90, 182, 274, 365, and

730 days. This permits a direct comparison of daily simulations with their monthly counterparts using the typical SPI-q naming scheme. This range is similar to the 1-60 month range considered by the US Drought Monitor (Svoboda et al., 2002; Xia et al., 2014), but we chose not to extend beyond 2 years (24 months). We chose to also include a 15 day window for completeness, to show behaviour under extreme conditions, and because some measures of flash drought rely on rapid decreases in water balance or soil moisture over 14 or 15 day periods (Christian et al., 2019; Lisonbee et al. 2022).


For daily and monthly experiments, we simulated a total of 10 million years using 20 repeated simulations of 500,000 years. The result was 3.65 billion total days and 0.12 billion months. Twenty repeated simulation was used to test uncertainty and

the impact of initial conditions, while also being conscientious of computational memory. Once it was determined that there were near imperceptible differences between statistical characteristics for the repeated simulations, the 20 simulations were combined to create the full 10 million year dataset. While this creates 19 small discontinuities at the interface between the 20 simulations, this effect was minimal when viewed across the 0.12-3.65 billion individual time steps

## 2.3 Temporal Autocorrelation at Case Study Sites

While the purpose of this study is primarily to determine the extreme behaviour from the Moving Average structure underlying many drought indices, it is illustrative to understand how close observed drought indices follow this idealized structure. To test this, we first derive SPI timeseries for the 1, 3, 6, 12, and 24 month moving window using daily data from the Global Soil Wetness Project Phase 3 (GSWP3) (Kim, 2017; Dirmeyer, et al. 2006). The GSWP3 forcing dataset is based on a the 20th Century Reanalysis dataset (Compo et al. 2011), with dynamical downscaling to maintain both high and low frequency signals and additional bias correction based on the GPCC (Meyer-Christoff et al. 2015) for precipitation. The GSWP3 dataset used here includes daily precipitation for the period 1901-2010 at a global resolution of 0.5°x0.5°. While a more comprehensive investigation is needed in the future to examine multiple variables, e.g. SPI, SPEI, SSI, SGI, at sites across the world, we have chosen to focus on 23 case study grid cells, aiming to span as many Koppen-Geiger climate zones (Peel et al. 2007) as feasible and to produce 3-5 sites in each of the six continents (Fig. A5). SPI was calculated following the stationary spline approach of Stagge and Sung (2022), by fitting a seasonally cyclic spline to the parameters of the Gamma distribution for positive precipitation and the logistic distribution for the probability of zero precipitation.

For each observed SPI time series, we then calculated the lagged correlation at 1 day increments using Pearson correlation. To compare observed autocorrelation with that expected from an idealized moving average process, we generated 1,000 independent replicates of 110 year daily time series following Eqs. 2 and 3 and calculated the lagged correlation for each using an identical approach to that used for the observed time series. The inner 95% percentile of all replicates was used to approximate the 95% confidence interval, while the mean showed the expected autocorrelation. Lagged autocorrelation from observed SPI was contrasted with that from the idealized moving average replicates to show how typical SPI series mimic the idealized series and to illustrate how much autocorrelation is caused by the moving average structure relative to climatological persistence for these example sites. We present four sites in North America in the text, but show all other sites as supplemental figures.

## 2.4 Annual Minima Analysis

Block annual minima were extracted from the simulated daily and monthly time series using an annual period. Calendar years, rather than water years, were used for ease of interpretation and because the synthetic SPI series does not distinguish seasonality.

To find a univariate probability distribution that reasonably approximates the simulated annual minima, we first compared the sample l-moment ratios with theoretical values for common univariate distributions (Hosking and Wallis, 1995; Peel et al., 2001). For the purposes of this analysis, the sign of annual minima was changed to better fit distributions typically designed for maxima extremes. This sign change is reasonable because, while seasonal precipitation or other measured variable may be skewed, the transformed SPI distribution is symmetrical, normally distributed about zero, and the simulated

SPI did not distinguish seasonality. L-moment plots were developed for the entire 10 million year simulated series and for each of the 20 subset simulations to estimate uncertainty around the l-moment estimate. In addition, we evaluated quantile-quantile (q-q) plots, showing the empirical quantiles from the sample compared with the theoretical quantiles determined from the candidate distribution.

Once an appropriate univariate distribution was found, we fit this distribution using both Maximum Likelihood Estimation (MLE) and l-moments, which have previously been shown to produce equivalent fits, particularly for large data sets (Beguería et al., 2014). Estimates across both methods were nearly equal. For all subsequent analyses we report l-moment estimates. Goodness of fit was verified using q-q plots and the Akiake Information Criterion (AIC) (Akaike, 1998; Cavanaugh and Neath, 2019). From the fitted distributions, exceedance probabilities were estimated, along with their

corresponding annual return periods for a range of SPI values using the l-moment estimates. Empirical estimates for several key exceedance probabilities were calculated as validation.

## 3 Results

### 3.1    Annual Minima Best Fit Distribution

To enable the development of continuous exceedance probability and return period curves, we sought to find a univariate

probability distribution that provided a good fit for the annual minima. L-moment ratios closely match the theoretical moments for the 3-parameter Generalized Normal distribution (Peel et al., 2001), also known as the 3-parameter Lognormal distribution (Fig. 2), indicating this distribution best fits the data. This holds true across all accumulation periods, regardless of whether one is using monthly (Fig. 2) or daily underlying data (Fig. A1). As the accumulation period increases towards 24 months, the annual minima appear increasingly like the normal, with l-skewness decreasing towards zero and l-kurtosis

approaching the theoretical value of 0.1226 for normally distributed data. The Generalized Normal Distribution is capable of representing the more skewed distributions for short accumulation periods and the more symmetrical distributions for longer accumulation periods, all while closely matching the simulated annual minima (Fig. 2). This extremely close fit supports the use of the Generalized Normal distribution to capture the magnitude of annual minima, as opposed to other potential distributions. This extremely close fit, for 10 million simulated observations, provides strong evidence. L-moment ratios

were extremely stable, with nearly imperceptible differences between the 20 simulations (Fig. A2). The interquartile range

(IQR) for l-skewness and l-kurtosis estimates from daily simulations were 0.0007-0.0015 and 0.0003-0.0007, respectively, amounting to uncertainty of 0.8-6% and 0.3-0.5%, measured by IQR/median.

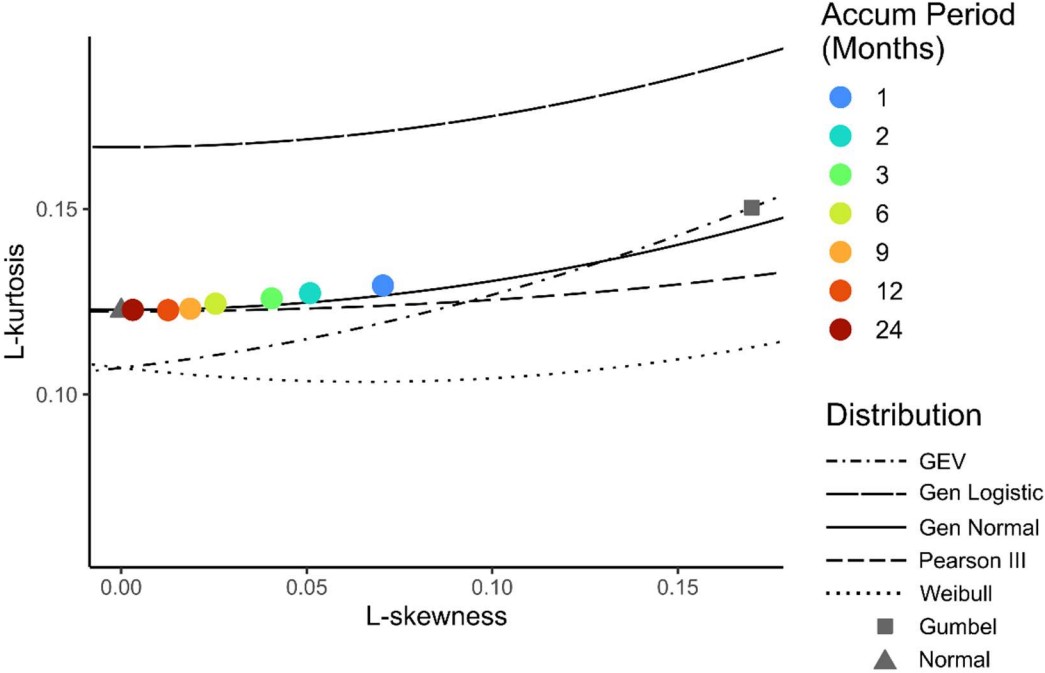

**Figure 2: L-moment ratios for annual extremes from monthly simulated series. Coloured points refer to fitted moments across varying accumulation periods, while lines correspond to theoretical distributions. Note, this figure shows distributions with a flipped sign. True skewness for annual minima is negative. An equivalent figure for daily simulations is shown in Appendix A (Fig. A1).**

The cumulative distribution function for the Generalized Normal distribution can be described by the standard normal distribution, $\Phi(Y)$, with an additional transformation, where Y is

$$
\begin{aligned}
Y &= -\kappa^{-1}\log(1 - \frac{\kappa(x-\xi)}{\alpha}) & \text{where } \kappa \neq 0 \\
Y &= \frac{x-\xi}{\alpha} & \text{where } \kappa = 0
\end{aligned}
\tag{4}
$$

$\xi$ is the location parameter, $\alpha$ is a scale parameter, and $\kappa$ is a shape parameter (Das, 2018; Hosking and Wallis, 1997). When $\kappa=0$, the distribution reverts back to the normal distribution with a mean of $\xi$ and standard deviation of $\alpha$. This distribution is equivalent to the 3-parameter lognormal distribution, or a normal distribution fit in natural log-space with parameters $\mu_{log}$, $\sigma_{log}$, and $\zeta$, corresponding to the location in log space, scale in log space, and a lower distribution bound (Das, 2018). For conversion between parameters of the two, one can use the following relationships: $\kappa = -\sigma_{log}$, $\alpha = \sigma_{log}e^{\mu_{log}}$, and $\xi = \zeta + e^{\mu_{log}}$.

The Generalized Normal distribution has been used in hydrology and meteorology studies (Basu and Srinivas, 2013; Das, 2018; Sangal and Biswas, 1970) for extreme value analysis.

Notably, the annual minima values do not converge towards the GEV distribution (Fig. 2), as might be expected for extreme values with temporal autocorrelation (Berman, 1964; Davis and Resnick, 1991; Hirtzel, 1985b; Leadbetter et al., 1983; Rootzen, 1986). The Berman theorem (Berman, 1964; Coles, 2001) states the maxima statistics of stationary Gaussian sequences with autocorrelations should converge towards a Gumbel distribution (GEV Type I) . We believe this deviation from expectation is due to the clustering of extremes, which violates Berman's theorem. This proposed explanation is expanded upon in the Discussion.

Deviations from the GEV distribution are most noticeable for longer accumulation periods, like the SPI-24, which has sample l-skew and l-kurtosis values of 0.0035 and 0.123, respectively (Fig. 2). These are nearly identical to theoretical values for the normal distribution (0 and 0.1226) and thus the Generalized Normal distribution, whereas the GEV distribution cannot produce distributions with zero skew (Fig. 2). While the deviation from the GEV distribution becomes smaller for shorter accumulation periods, it is notable how closely the empirical l-moments from the simulations follow the Generalized Normal distribution (Fig. 2).

Quantile-quantile plots were used to further verify fitting skill, with all simulated extremes falling neatly along the distribution, with only slight deviation at the most extreme values (return periods > 1 million years) and no consistent patterns of bias (Fig. A3). The Generalized Normal distribution therefore accurately reproduces empirical quantiles, with little noticeable bias even at the extremes. These strong fits appear similarly accurate for short and long accumulation periods, though AIC values slightly increase (become worse) for longer accumulation periods (Fig. A4a). The AIC confirms that the Generalized Normal distribution produces a better fit than the GEV across all moving average lengths, illustrated by a lower (negative difference) in AIC (Fig. A4b). Based on all of this evidence, all subsequent analyses are therefore based on the Generalized Normal distribution, except where empirical estimates are used as a validation.

## 3.2    Observed vs Theoretical Autocorrelation

The pattern of temporal autocorrelation for real-world SPI time series follows the pattern from the idealized moving average time series, with a mostly linear decrease towards zero at the moving window length, followed by fluctuations around zero (Fig 3 and Supplemental Figs. A6-10). This implies that the structural persistence, occurring due to the moving average, is more important than climatological persistence in many locations and climate regions. The first row of Fig 3 shows the first 20 of the 1,000 replicates used to generate the 95% interval. The red, observed line generally remains within this 95% interval for climates ranging from cold, hot summer (Winnipeg, group Dfb), to temperate (Columbus, group Cfa), to tropical monsoon (Miami, group Am), and hot desert (Tucson, group BSh). This pattern holds for all other case study sites analysed

outside of North America (Supplemental Figs. A6-10). Temporal autocorrelation beyond the limits shown here continues to fluctuate around zero, but generally remains within the 95% interval. Where there are deviations from the theoretical persistence, this could be evidence of randomness or of some climatological persistence. This can be investigated more

thoroughly in future studies (Section 4.1).

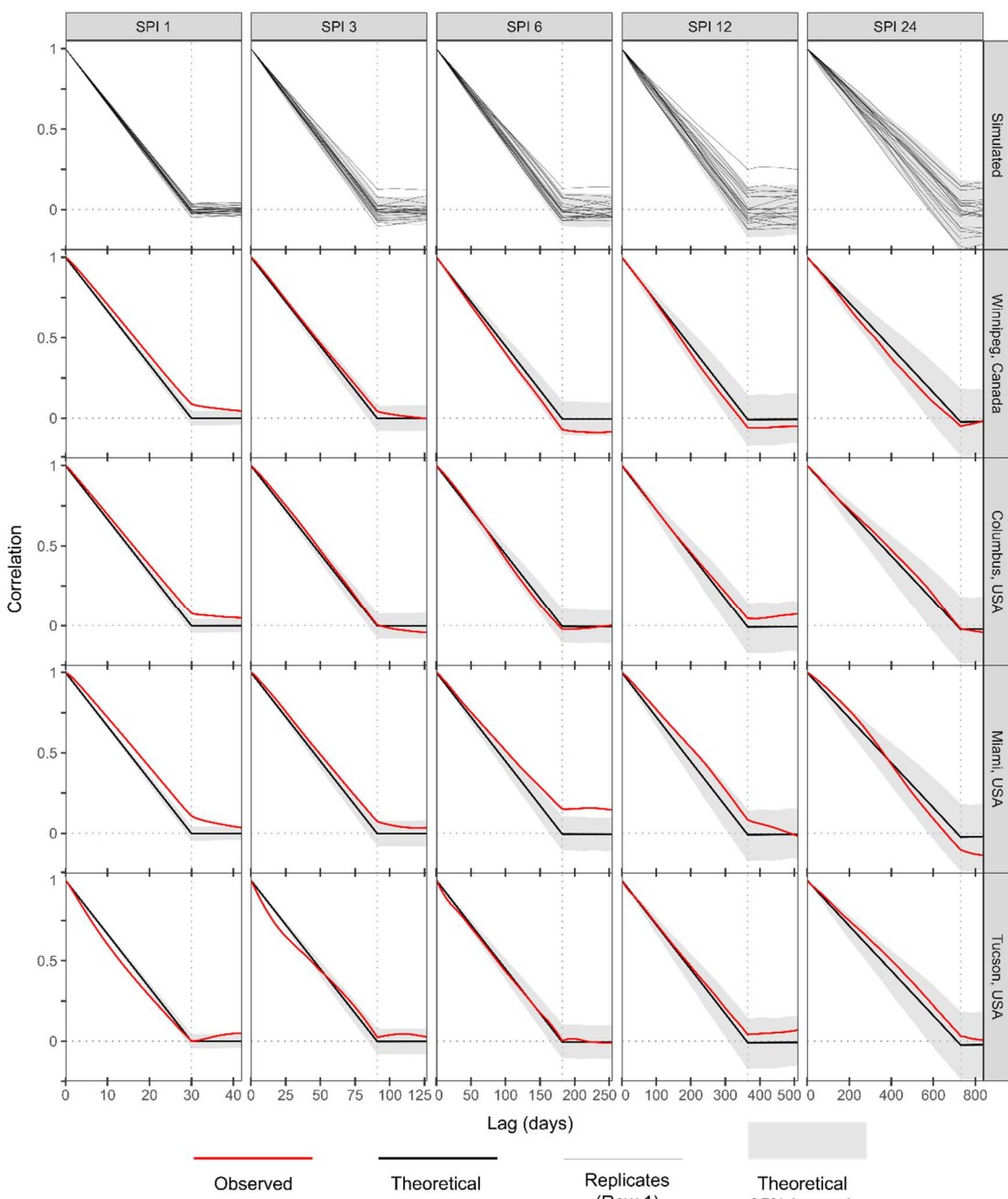

**Figure 3: Lagged correlation for the SPI-1, 3, 6, 12, and 24 moving windows. The first row shows 20 replicates from random simulation, following Eqs. 2 and 3, in light grey. Subsequent rows show four North American grid cells ranging in climate zones from coldest (Winnipeg) to warmest (Tucson). Red lines show observed temporal autocorrelation. Black lines show the expected (mean) temporal autocorrelation from 1,000 replicate simulations, while the grey regions show the 95% interval from these replicates. Dotted lines show the moving window length. Note, each subfigure has a different x-axis to better focus on the moving window.**

**3.3      Annual Extreme Values**

Using the fitted Generalized Normal distribution, we explored the distribution and return periods for annual minima of the idealized Moving Average time series described by Eqs. 2 and 3. For longer accumulation periods, the distribution of annual minima becomes less skewed, with a shift in the mean towards zero (Figs. 4 and S2). Conversely, short accumulation periods have more skew and are shifted towards the negative (more extreme). For daily data, the distribution mean increases from -

2.53 for a one month accumulation period (SPI-1) to -0.77 for a 24 month period (SPI-24) (Figs. 5 and S2). For monthly data, this shift in the mean value for the annual minima increases from -1.63 to -0.61 for the SPI-1 and SPI-24 respectively. Differences between the daily and monthly resolutions are discussed in the next section.

Concurrent with an increase in the distribution mean for the SPI annual minima, distributions transition from skewed left for

short accumulation periods towards more normally distributed (negligible skew) for long accumulation periods (Figs. 5 and S4). Variance also increases with increased accumulation period. All distribution parameters and moments are presented in Appendix A (Figs. S5 and S6, Table S1).

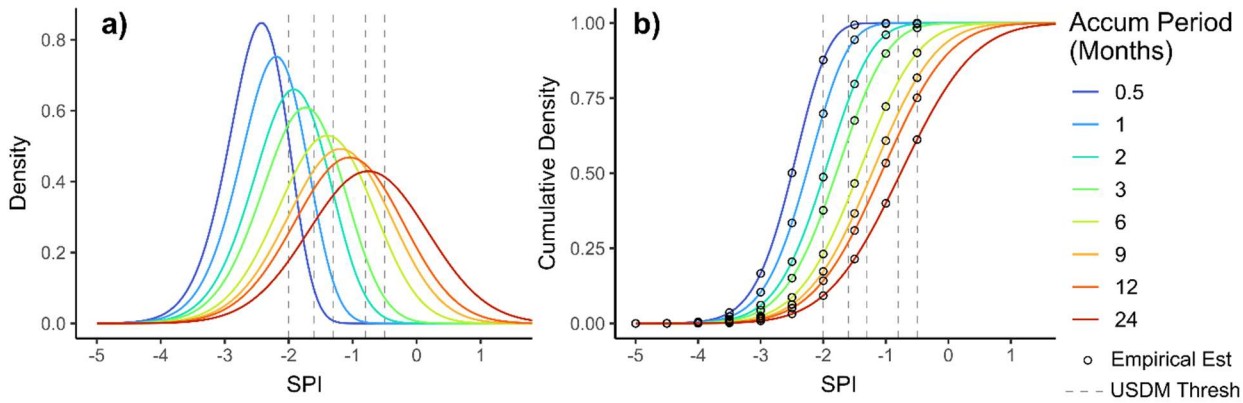


**Figure 4: Annual minima (a) distribution and (b) cumulative probability density for daily sequences of varied accumulation periods, indicated by colour. Colours are identical to Fig. 2. Vertical grey lines correspond to US Drought Monitor thresholds for D0-D4 (-0.5, -0.8, -1.3, -1.6, and -2.0). Open points represent empirical estimates directly from simulation.**

The aforementioned distribution changes due to accumulation period produce differences in the probability of annual threshold exceedances and their associated return period (Figs. 4 and 5). In Fig. 5, lines represent the probability of threshold exceedances derived from the fitted distribution, while vertical lines correspond to USDM thresholds. If one was to focus on the D4 exceptional drought threshold (SPI < -2.0) for a daily time series, the annual probability of a single SPI-3 exceedance

of this threshold is 37.7%, a return period of 2.65 years, while this probability decreases to 9.27% for the SPI-24,

corresponding to once every 10.79 years (Fig. 5a, Table 1). The discrepancy becomes even greater for extremely short accumulation periods. For example, the probability of at least one value below -2 for the SPI-0.5 (15 day period) is 87.6% (return period of 1.14 years). In other words, a drought agency would declare a D4 drought every year if monitoring the SPI-0.5, but only once a decade if monitoring the SPI-24.

Another way to interpret Fig. 5 is to make a horizontal comparison. The 2 year return period should be exceeded once every other year when measured a sufficiently long record. The threshold associated with this relatively commonplace occurrence varies from -2.5, considered an Extreme (D3) drought for the SPI-0.5 to -0.764, considered only a moderate (D1) drought) for the SPI-24 (Fig. 5a). The idea of experiencing an "Extreme" short drought (0.5 month accumulation period) at least once every other year may be challenging for interpretation by the public. Again, this difference is solely due to the structural

behaviour of the SPI's moving average and is important to understand when comparing SPI extreme occurrences from different accumulation periods.

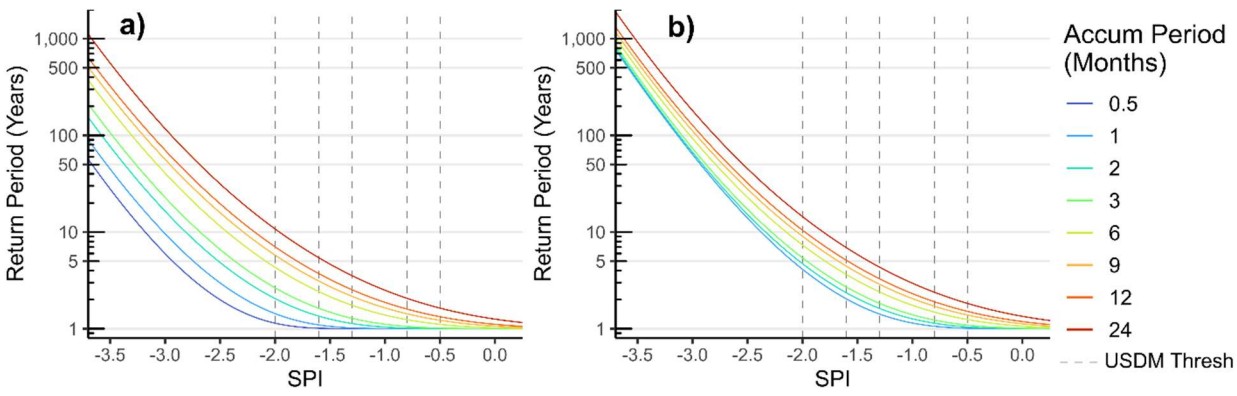

**Figure 5: Return periods for (a) daily and (b) monthly sequences with accumulation periods indicated by colour. Colours are**
**identical to Figs. 2 and 3. Vertical grey lines correspond to US Drought Monitor thresholds, identical to Fig. 4.**

Table 1: Annual return periods (in years) using daily simulation, for various SPI thresholds and accumulation periods.

| Accumulation Period | | Return Period (Years) for SPI Threshold | | | | | | | | |
|---|---|---|---|---|---|---|---|---|---|---|
| Months | Days | -4 | -3.5 | -3 | -2.5 | -2 | -1.5 | -1 | -0.5 | 0 |
| 0.5 | 15 | 185 | 28.2 | 5.97 | 1.99 | 1.14 | 1.01 | 1.00 | 1.00 | 1.00 |
| 1 | 30 | 289 | 45.4 | 9.61 | 2.97 | 1.43 | 1.06 | 1.00 | 1.00 | 1.00 |
| 2 | 60 | 473 | 76.8 | 16.4 | 4.83 | 2.05 | 1.26 | 1.04 | 1.00 | 1.00 |

| | | -4 | -3.5 | -3 | -2.5 | -2 | -1.5 | -1 | -0.5 | 0 |
|---|---|---|---|---|---|---|---|---|---|---|
| 3 | 90 | 659 | 107 | 22.9 | 6.60 | 2.65 | 1.48 | 1.11 | 1.02 | 1.00 |
| 6 | 182 | 1060 | 182 | 40.1 | 11.5 | 4.32 | 2.15 | 1.39 | 1.11 | 1.02 |
| 9 | 274 | 1440 | 251 | 55.6 | 15.8 | 5.78 | 2.73 | 1.65 | 1.22 | 1.06 |
| 12 | 365 | 1820 | 315 | 69.4 | 19.6 | 7.04 | 3.24 | 1.88 | 1.33 | 1.11 |
| 24 | 730 | 3520 | 569 | 118 | 31.6 | 10.8 | 4.65 | 2.50 | 1.63 | 1.26 |

Return periods for extreme SPI values rapidly increase beyond -3.5 (Fig. A7). Return periods from daily data for SPI= -4 range from 185 to 3,520 years, depending on the accumulation period, and from 14,600 to 291,000 years for SPI equal to -5. These values differ from those generated by Stagge et al. (2016), which focused on recurrence within a given day of the year rather than on annual minima. However, both convey the same concern, that using the SPI or other normalized drought indices to quantify tail behaviour at such extremely low probabilities is dubious given common record lengths of 100 years

or less.

The difference in return period for annual SPI minima from a theoretical time series noted here is solely due to the structural persistence caused by the size of the moving window. In case study sites using the SPI, structural persistence appears to play by far the predominant role in temporal autocorrelation (Fig. 3) The effect of structural persistence on annual minima can be

partially explained because shorter accumulation periods necessarily have larger innovations, $\sqrt{q}/q$, leading to more erratic behaviour, while maintaining the same overall standard normal distribution for each day of the year. As accumulation periods become larger, the moving window averages more individual days or months, requiring more sustained anomalies to produce extreme values. For the longest accumulation periods, where the moving average window becomes longer than a year, the resulting time series slowly transitions from positive to negative or vice versa over the course of multiple years,

thereby producing even occasional years in which the annual minima is greater than zero (Figs. 4 and 5).

**Table 2: Annual return periods (in years) using monthly simulation, for various SPI thresholds and accumulation periods.**

| Accumulation Period | Return Period (Years) for SPI Threshold | | | | | | | | |
|---|---|---|---|---|---|---|---|---|---|
| Months | -4 | -3.5 | -3 | -2.5 | -2 | -1.5 | -1 | -0.5 | 0 |
| 1 | 2660 | 378 | 64.5 | 14 | 4.11 | 1.77 | 1.15 | 1.01 | 1.00 |
| 2 | 2830 | 391 | 67.6 | 15.3 | 4.7 | 2.05 | 1.27 | 1.05 | 1.00 |
| 3 | 2950 | 415 | 73.9 | 17.2 | 5.4 | 2.34 | 1.39 | 1.09 | 1.01 |

| 6 | 3290 | 489 | 92.7 | 22.8 | 7.34 | 3.13 | 1.75 | 1.24 | 1.06 |
|----|------|-----|------|------|------|------|------|------|------|
| 9 | 3260 | 526 | 107 | 27.4 | 9.02 | 3.81 | 2.05 | 1.39 | 1.12 |
| 12 | 3840 | 613 | 124 | 31.8 | 10.4 | 4.37 | 2.31 | 1.51 | 1.19 |
| 24 | 6020 | 918 | 180 | 45.2 | 14.5 | 5.85 | 2.96 | 1.83 | 1.34 |

**3.4 Effect of Temporal Resolution**

The temporal resolution of the underlying data (monthly or daily) has a strong impact on the annual minima of the simulated SPI. Using daily data shifts the distribution of the annual minima to become more extreme (more negative) across all accumulation periods, though the effect is strongest for short accumulation periods (Figs. 6 and 7). In turn, this makes return periods for monthly resolution data longer, even when considering the same threshold (Fig. 7). For example, the SPI-3 is

likely to exceed -2 at least once every 2.65 years (p=0.378) when using daily data, but only exceed this threshold once every 5.40 years (p=0.185) when using monthly data (Tables 1 and 2). For the SPI-12, return periods for daily and monthly series become 7.04 and 10.44 years, respectively for the -2 threshold. This is tied to the number of random samples (12 vs 365) and the increased likelihood of a random extreme outlier, despite moving windows of equal lengths.

There is little difference between the higher moments (variance, skewness, kurtosis) when comparing the distribution of annual extremes generated from daily or monthly data (Fig. A2), despite the moving window representing the same proportion of the year. Seemingly the only difference is in the distribution mean, which is shifted to the more extreme (lower) when using daily data (Fig. A5). Because monthly and daily SPI data are often used interchangeably, the effect of temporal resolution on annual minima is important to acknowledge by practitioners and drought monitoring agencies, as it is

purely an artefact of calculation procedure and not the climate.

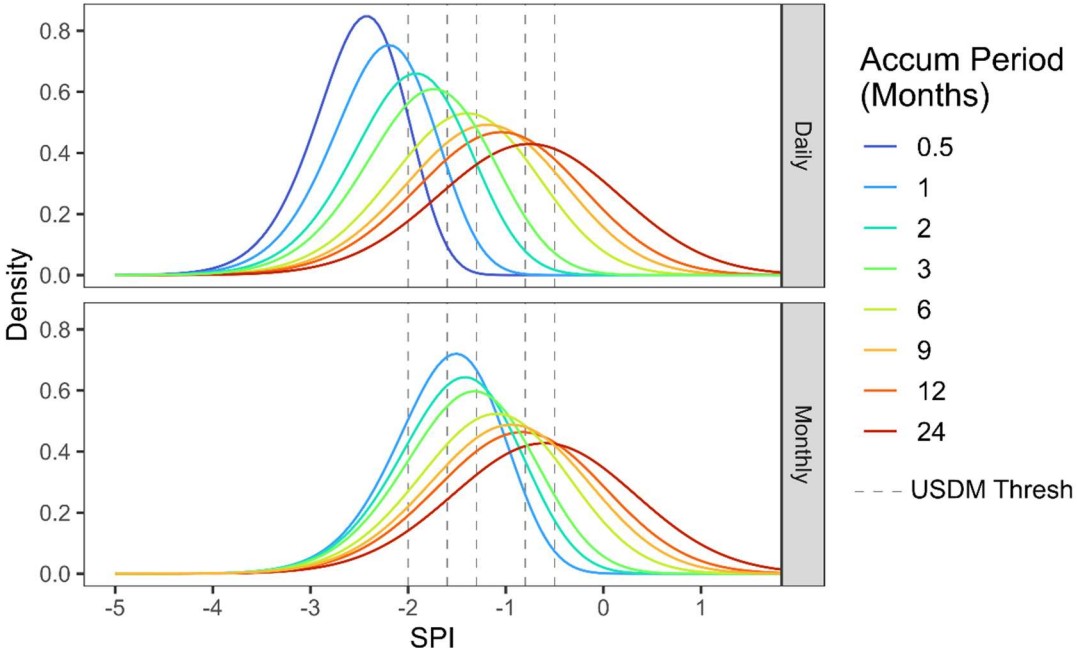

**Figure 6: Fitted distribution of annual minima from daily (upper) and monthly (lower) time series. Colours and vertical lines are identical to Figs. 4 and 5.**


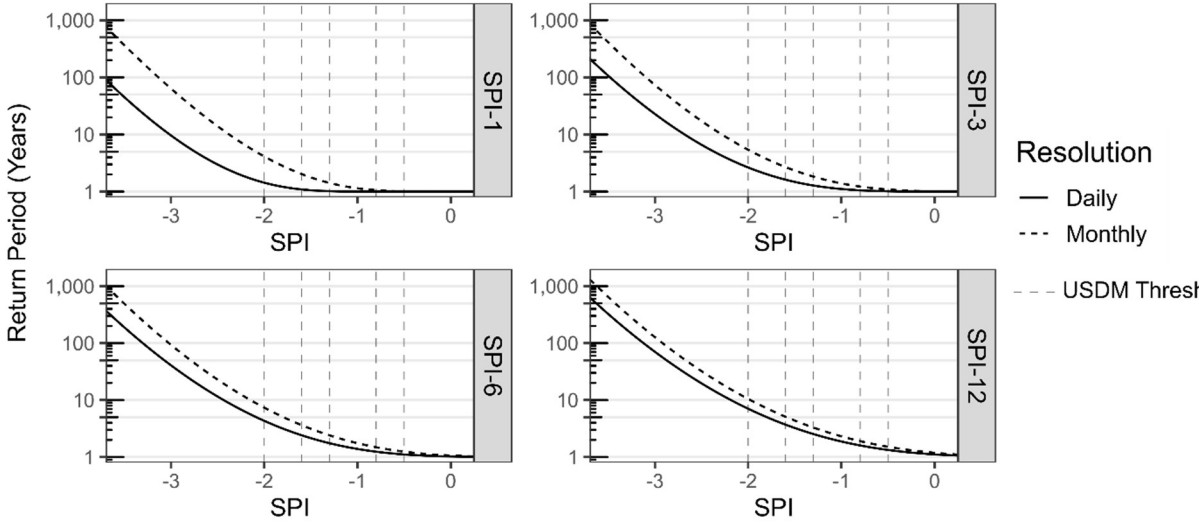

**Figure 7: Return period comparison between daily (solid) and monthly (dotted) underlying data, where SPI value is presented on the x-axis and return period is shown on the y-axis in a log scale. USDM thresholds are shown, as in previous figures.**

## 4 Discussion

### 4.1 Structural Persistence vs Real World Persistence

This study is, to our knowledge, the first attempt to quantify the return frequency of annual minima from a standardized drought index following a moving average structure. To accomplish this, we defined the behaviour of a stochastic model that mimics the moving average of the SPI (Eq. 2 and 3). In practice, the SPI and other drought indices can deviate from this general model in several ways, described next. Therefore, these results represent annual minima return periods for a highly idealized system as a bounding case, considering only structural persistence and Gaussian (symmetrical) innovations.

Normalized drought indices derived from gauge data can be subject to climatological persistence in addition to structural persistence. Structural persistence is caused by the moving window structure, whereas climatological persistence is caused by climatological patterns, including frontal systems at shorter time scales and teleconnections like El Niño or the North Atlantic Oscillation at longer time scales. Another potential deviation from our theoretical SPI model is the seasonal regime, which may cause certain seasons to be more strongly correlated or to undergo rapid overturning of conditions. A final deviation from the real-world is our assumption of symmetrical innovations (Eq. 2), which may not always hold true. This is particularly relevant for regions with low absolute precipitation, where individual large storm events may produce more extreme positive innovations than for negative innovations.

Despite the potential for deviations from the idealized Moving Average model that is the focus of this study, we found that most observed SPI series follow the expected structural persistence (Fig. 3). This finding suggests that these results are relevant for most SPI series, though one should check the persistence structure if they are to use this approach. Though the 23 global sites tested here do not show major climatological persistence, there may be regions with notable deviations from the assumed structural persistence created by the moving window. Testing of the persistence structure for SPI, SPEI, and other moving average drought indices on a global, gridded scale would be needed to fully confirm this finding and should be addressed in future research.

### 4.2 Theoretical Basis Against GEV

Extreme value theory, based on the work of Berman (1964), suggests that for time series with high autocorrelation, including the moving average, the annual minima should asymptotically converge towards a member of the GEV distribution, namely the Gumbel distribution (Davis and Resnick, 1988; Eichner et al., 2006; Hirtzel, 1985b; Husler, 1990; Leadbetter et al., 1983; Rootzen, 1986). Notably, our simulations deviate from this, instead converging towards the Generalized Normal Distribution (Fig. 2 and A1).

One potential explanation for this deviation is that the Moving Average structure imparts extremal clustering (Coles, 2001; Moloney et al., 2019) to the time series, which violates the assumptions underlying the Berman theorem. Extremal clustering, the tendency for a time series to cluster at extreme levels, is quantified by the extremal index, $\theta$ (Coles, 2001; Moloney et al., 2019). Clustering of extremes differs from temporal autocorrelation as it only considers the clustering for events above a given extreme threshold (Auld and Papastathopoulos, 2021; Lindgren et al., 1983; Moloney et al., 2019). The

extremal index, $\theta$, ranges from 0 to 1, representing completely independent extremes ($\theta = 1$) and progressively smaller values of $\theta$ representing larger degrees of extremal clustering. The Berman theorem (Berman, 1964) and its subsequent derivations for Moving Average sequences (Rootzen, 1986) are predicated on minimal extremal clustering. However, the moving averages used in normalized drought indices have increasing levels of extremal clustering for longer accumulation periods (Fig. A15, lower $\theta$ with higher accumulation). These levels of extremal clustering are still relatively minor, but may

partially explain the deviation from the GEV towards the Generalized Normal distribution, especially for longer accumulation periods (Fig. 2). More research would be required to confirm this hypothesized cause.

From a physical perspective, it is logical that moving average drought indices have increased extremal clustering, particularly for long accumulation periods because the longer moving window approaches or surpasses the size of the annual

block. Long moving average windows produce relatively small incremental changes, which makes it unlikely to reach extreme minima in a given year without the preceding year already being quite low. This, in addition to smearing drought events across neighbouring calendar years, leads to greater levels of extremal clustering and lower values of $\theta$ (Fig. A15). Eichner et al. (2006) noted similar behaviour for annual block maxima derived from an autocorrelated series, finding that distributions become more normally distributed with higher degrees of correlation.

**5 Conclusions**

This study represents is an advancement for the understanding of annual extremes derived from Moving Average time series, of which normalized drought indices like the SPI are an important type. The SPI and other normalized drought indices are used by drought monitoring agencies throughout the world to quantify the relative severity of droughts and to classify conditions into discrete drought states. Because of the importance placed on these indices, this study explored the behaviour

of a theoretical, idealized moving window sequence with respect to annual minima. The major advances shown here can be viewed from two perspectives: an improved theoretical understanding of moving window sequences and practical findings for the application of drought indices by drought monitoring agencies when.

From a theoretical perspective, this study presents a stochastic model to simulate a Moving Average process where the

moving average window is proportionally large (5-200%) relative to the year (Eq. 2 and 3). This produced the first, to our knowledge, explicit quantification of annual extreme exceedances from such a sequence. We showed that the distribution of

annual minima follow a Generalized Normal distribution, rather than the GEV distribution, which was the initial expectation from extreme value theory. This deviation is likely due to extremal clustering.

From an applied perspective, this study provides the expected annual return periods for the SPI or related drought indices with common accumulation periods ranging from 1 to 24 months (Fig. 5, Tables 1 and 2). We show that the likelihood of exceeding an SPI threshold in a given year decreases (annual return period increases) as the accumulation period increases. The corollary of this finding is also true; the SPI threshold associated with a given annual return period becomes less extreme (closer to zero) for indices with longer accumulation periods. Practitioners have implicitly understood this

relationship, even from the first definition of the SPI (McKee et al., 1993), which included a figure showing the number of unique droughts per 100 years decreasing with longer accumulation periods for a gauge in Ft. Collins, CO. Likewise, the European Drought Observatory's Combined Drought Index uses a more extreme threshold for short accumulation periods (SPI1 < -2) than for longer accumulation periods (SPI3 < -1) when classifying drought regions, in line with our findings that suggest these thresholds should have similar annual return periods (1-1.5 years) for daily data (Fig. 5, Table 1). Despite an

implicit understanding by drought practitioners, this study is the first to explicitly calculate the return period for a theoretical normalized drought index using a moving average. Drought managers can use this knowledge to better interpret exceedances of the SPI or other drought metrics.

A second practical finding is that annual minima from a normalized drought index (e.g. the SPI or SPEI) depend on whether

one uses daily or monthly data, even for the same accumulation period. Drought practitioners have largely understood that daily data is more noisy, and thus more subject to single day deviations towards extremes, but this effect has not been quantified explicitly to date. Further, researchers tend to use daily or monthly resolution data interchangeably if the accumulation periods are equivalent, which we have shown here to produce different results when viewed as annual exceedances.


We therefore propose several recommendations. The first is a general recommendation for users of the SPI and related drought indices to be careful with language and thoughtful about interpretation when using a normalized drought index to determine whether a given event is particularly extreme. Our goal is to clarify the difference between the probability of the SPI exceeding -2 on a specific day from the probability of exceeding -2 in a given year (commonly called return period). The

former is the definition of the SPI, whereas the latter is covered in this study. This distinction should be clear when considering that the likelihood of SPI < -2 on any given day is 2.3%, but the likelihood of experiencing SPI < -2 in a year ranges from 6.9% to 87.7%, depending on the accumulation period and temporal resolution. This leads to the second recommendation, which is that practitioners should exercise caution when comparing the likelihood or severity of a particularly extreme SPI value using a short accumulation period with one using a long accumulation period. It is most

appropriate to compare an index with itself and if one must make cross-index comparisons, there may be a need to use different thresholds, as is done by the European Drought Observatory's Combined Drought Index.

Our final recommendation is for more research to explore the findings shown here. For example, more research should explore the degree to which climatological persistence and seasonality affect temporal autocorrelation across a range of

climate regions and drought indices, expanding on the example sites and SPI tested in Section 3.2. The theoretical case developed here should act as a baseline under an idealized moving average time series. But, deviations from this may be illustrative. While this result was meant as the simplest case, using the block annual minima for an idealized time series, future research could explore more drought specific indices like duration.


**Appendix A. Additional Figures**

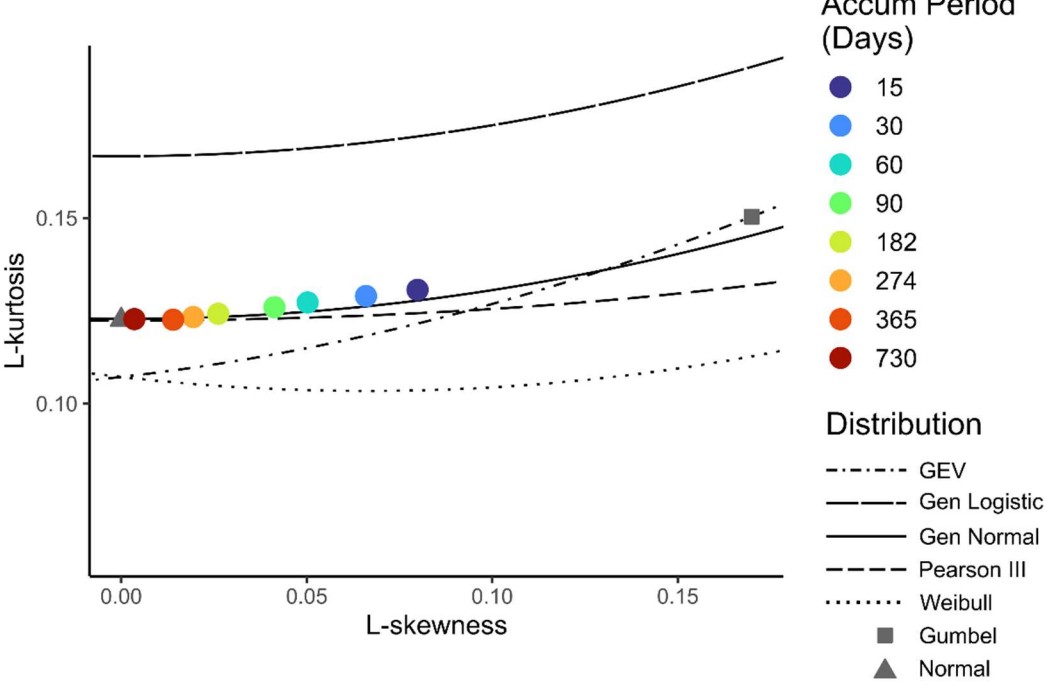


**Figure A1: L-moment ratios for annual extremes from daily simulated series. Coloured points refer to fitted moments across varying accumulation periods, while lines correspond to theoretical distributions. Note, this figure shows distributions with a flipped sign. True skewness for annual minima is negative.**


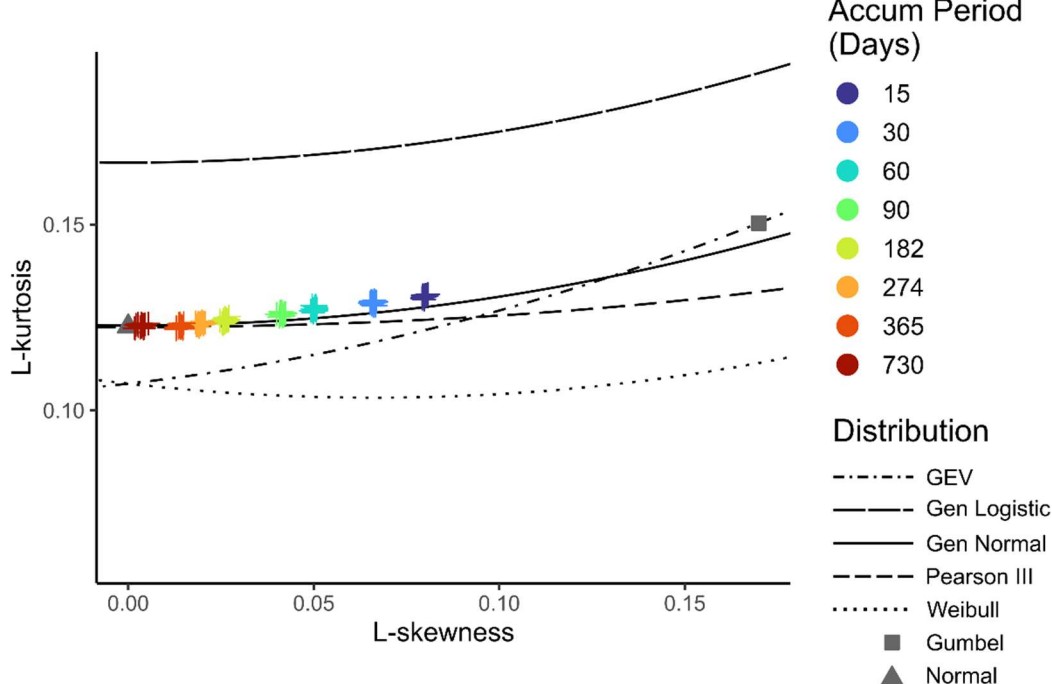

**Figure A2: L-moment ratios for annual extremes from daily simulated series. Each of the 20 replicates are shown as unique crosses (+). Colours refer to accumulation periods, while lines correspond to theoretical distributions. Note, this figure shows distributions with a flipped sign. True skewness for annual minima is negative.**


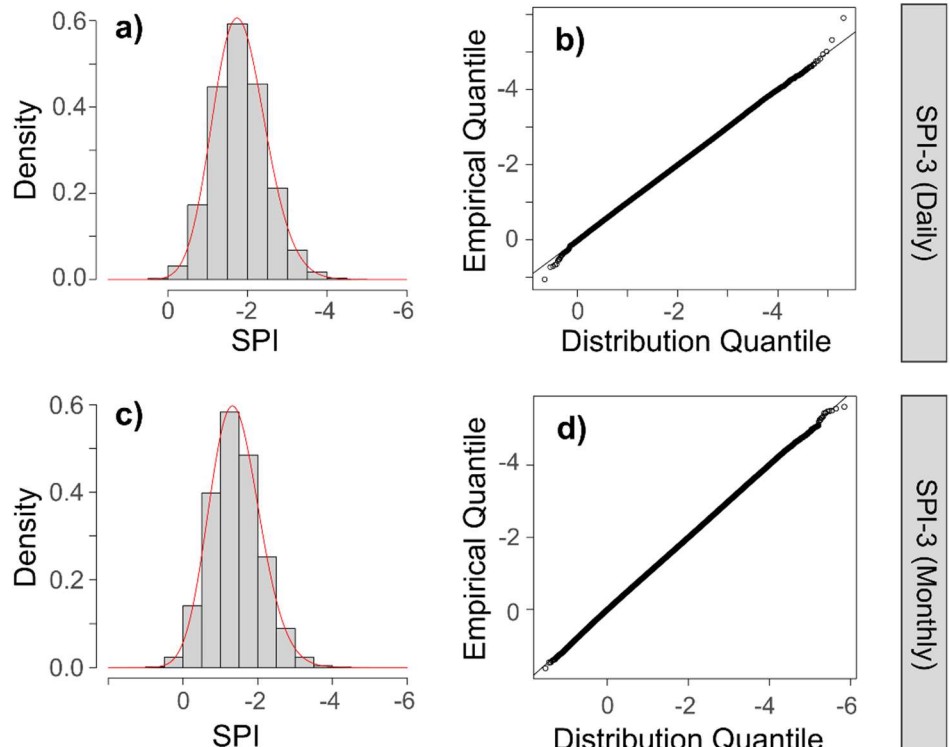

**Figure A3: Example distribution fit for the SPI-3 using (a-b) daily and (c-d) monthly data. On the left (subfigures a and c) empirical density is shown as a grey histogram, while the fitted Generalized Normal distribution is shown in red. The right (subfigures b and d) shows a quantile-quantile plot comparing empirical to fitted quantiles.**


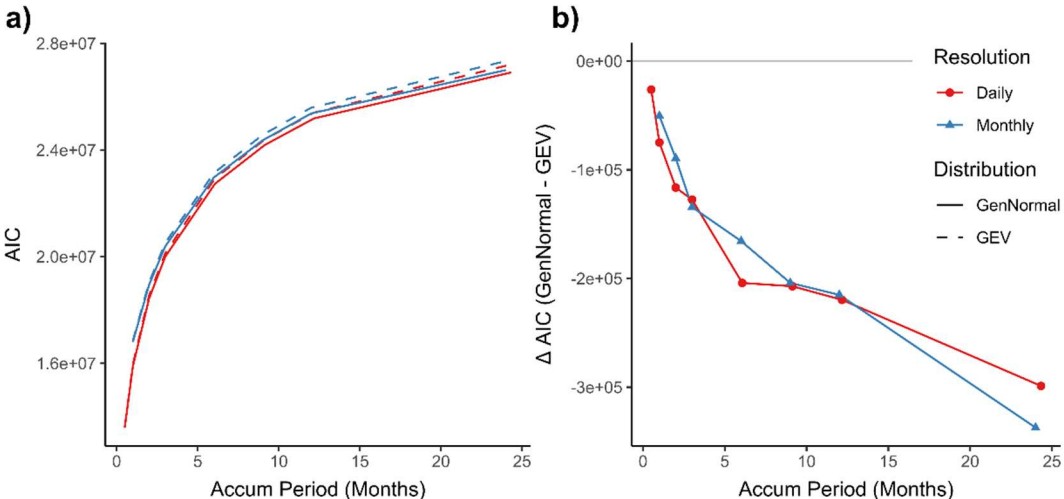


**Figure A4: Comparison of Akaike information criterion (AIC) between the Generalized Normal and GEV distributions. This is shown as (a) raw AIC values, where lower values represent better fits, and (b) the difference between AIC for the Generalized Normal and GEV distributions, where negative values suggest the Generalized Normal distribution fits better. Colours refer to the distribution in both figures, while the line style (solid vs dashed) refers to the distribution.**


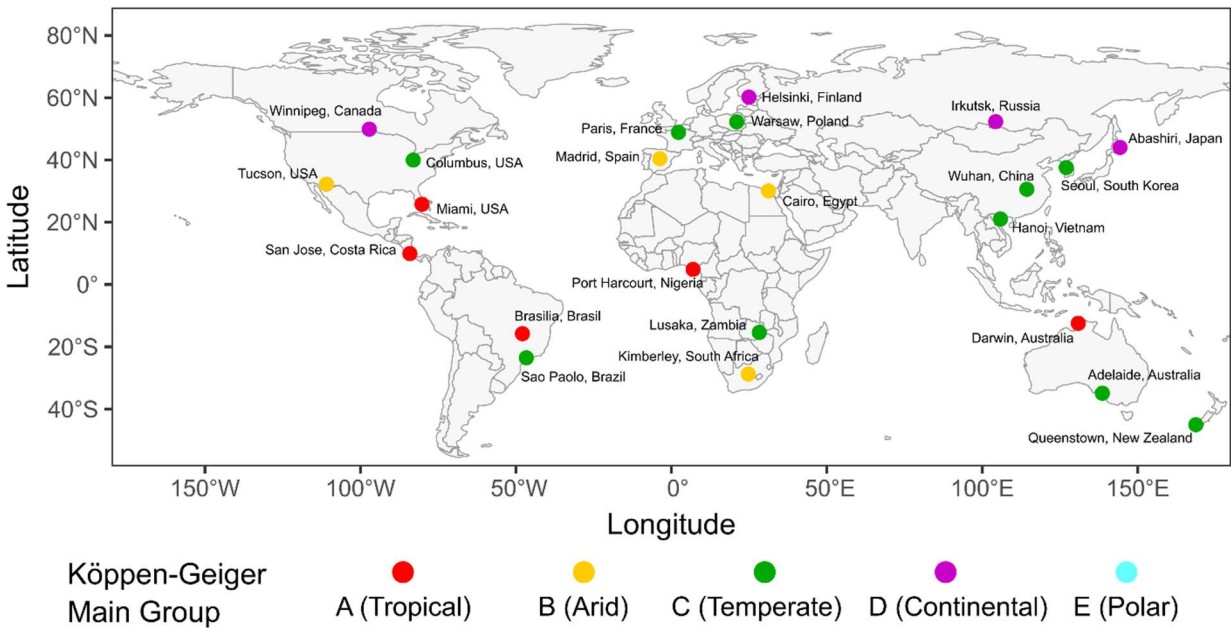


**Figure A5: Location of case study grid cells used to compare observed SPI temporal autocorrelation with theoretical autocorrelation.**


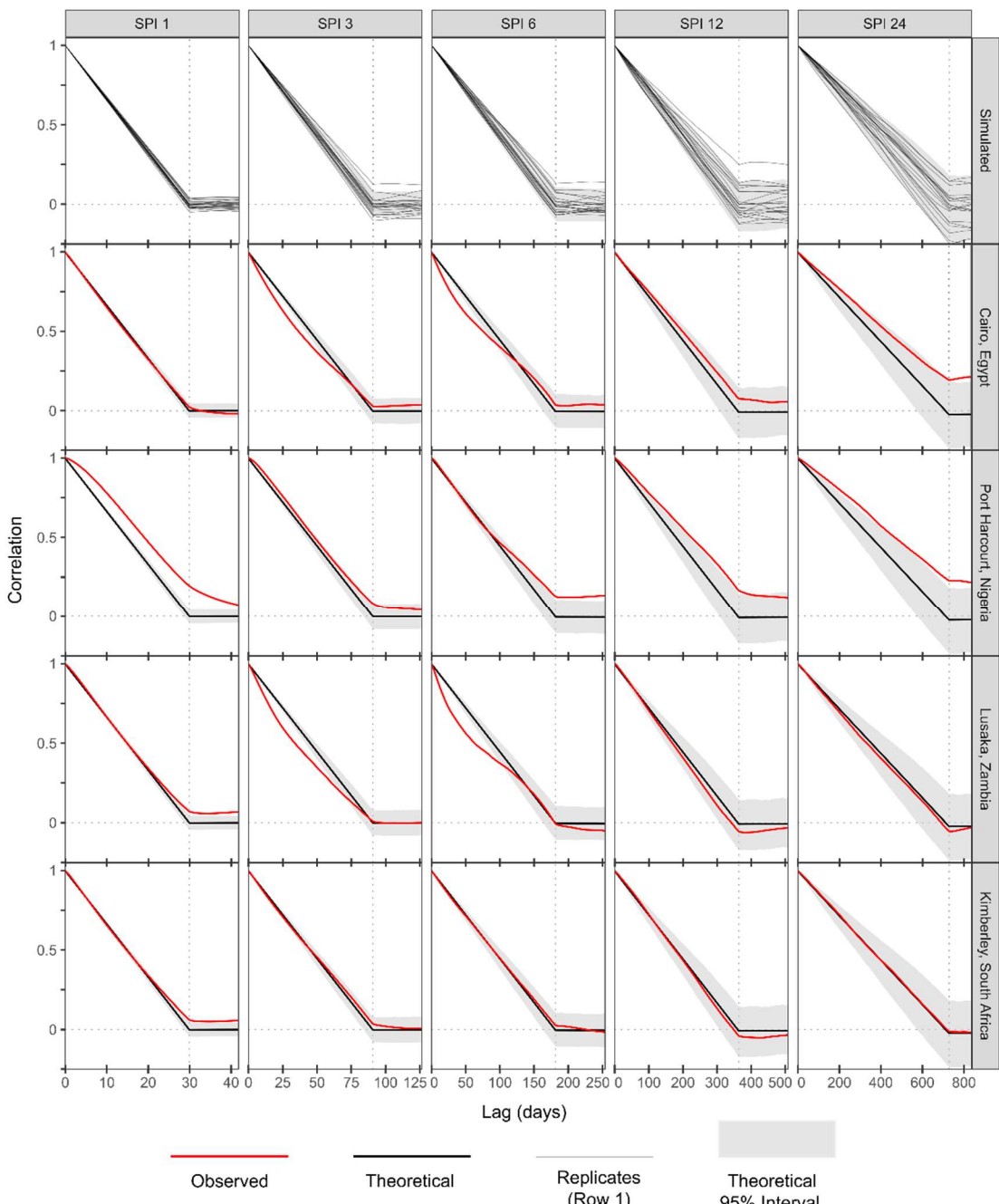

**Figure A6: Identical format to Fig 3 for representative grid cells in Africa. Lagged correlation for the SPI-1, 3, 6, 12, and 24 moving windows. The first row shows 20 replicates from random simulation in light grey, while subsequent rows contrast observed autocorrelation (red) with theoretical (grey interval with black line).**


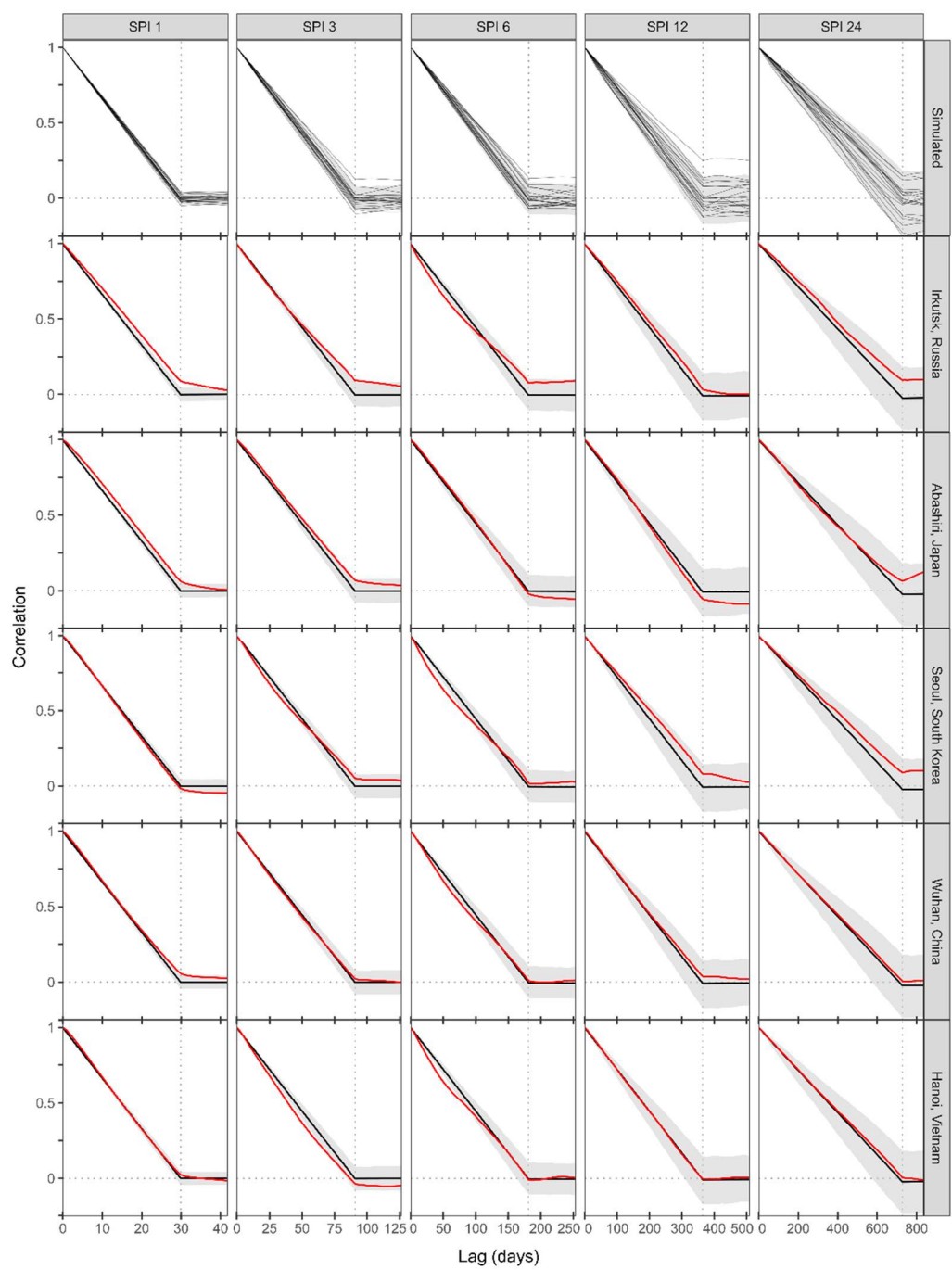

**Figure A7: Identical format to Fig 3 for representative grid cells in Africa. Lagged correlation for the SPI-1, 3, 6, 12, and 24 moving windows. The first row shows 20 replicates from random simulation in light grey, while subsequent rows contrast observed autocorrelation (red) with theoretical (grey interval with black line).**

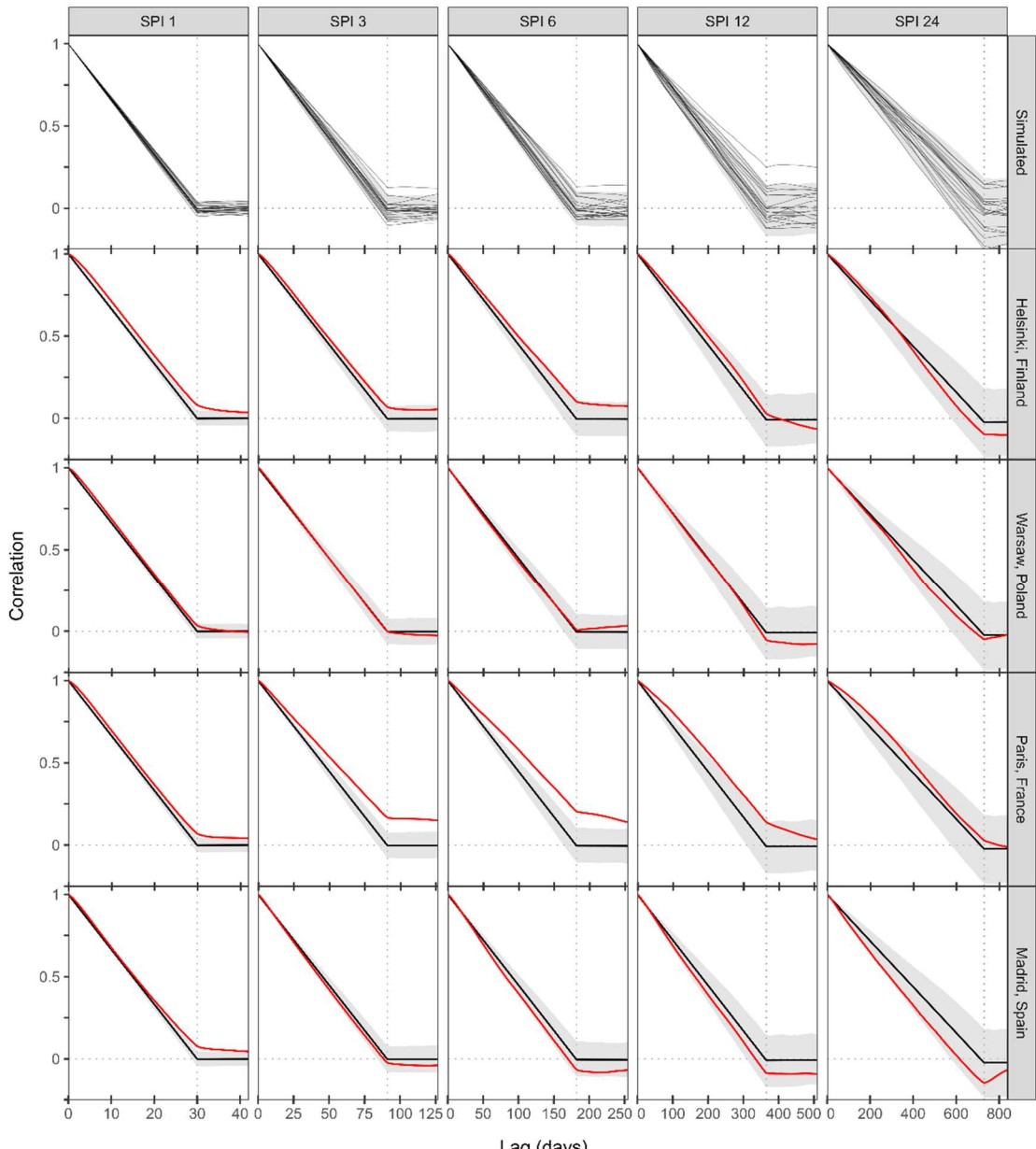

**Figure A8: Identical format to Fig 3 for representative grid cells in Europe. Lagged correlation for the SPI-1, 3, 6, 12, and 24 moving windows. The first row shows 20 replicates from random simulation in light grey, while subsequent rows contrast observed autocorrelation (red) with theoretical (grey interval with black line).**

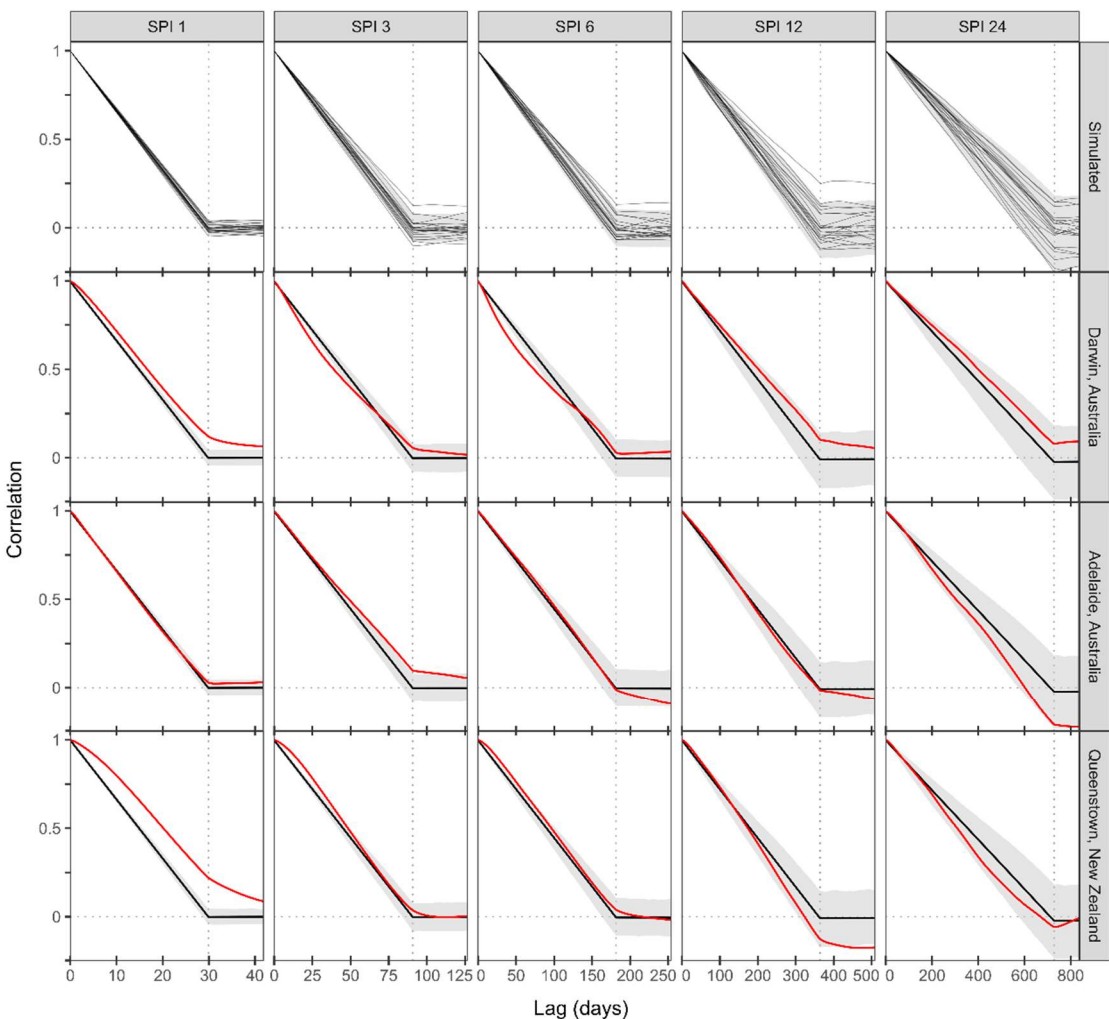

**Figure A9: Identical format to Fig 3 for representative grid cells in Oceania. Lagged correlation for the SPI-1, 3, 6, 12, and 24 moving windows. The first row shows 20 replicates from random simulation in light grey, while subsequent rows contrast observed autocorrelation (red) with theoretical (grey interval with black line).**


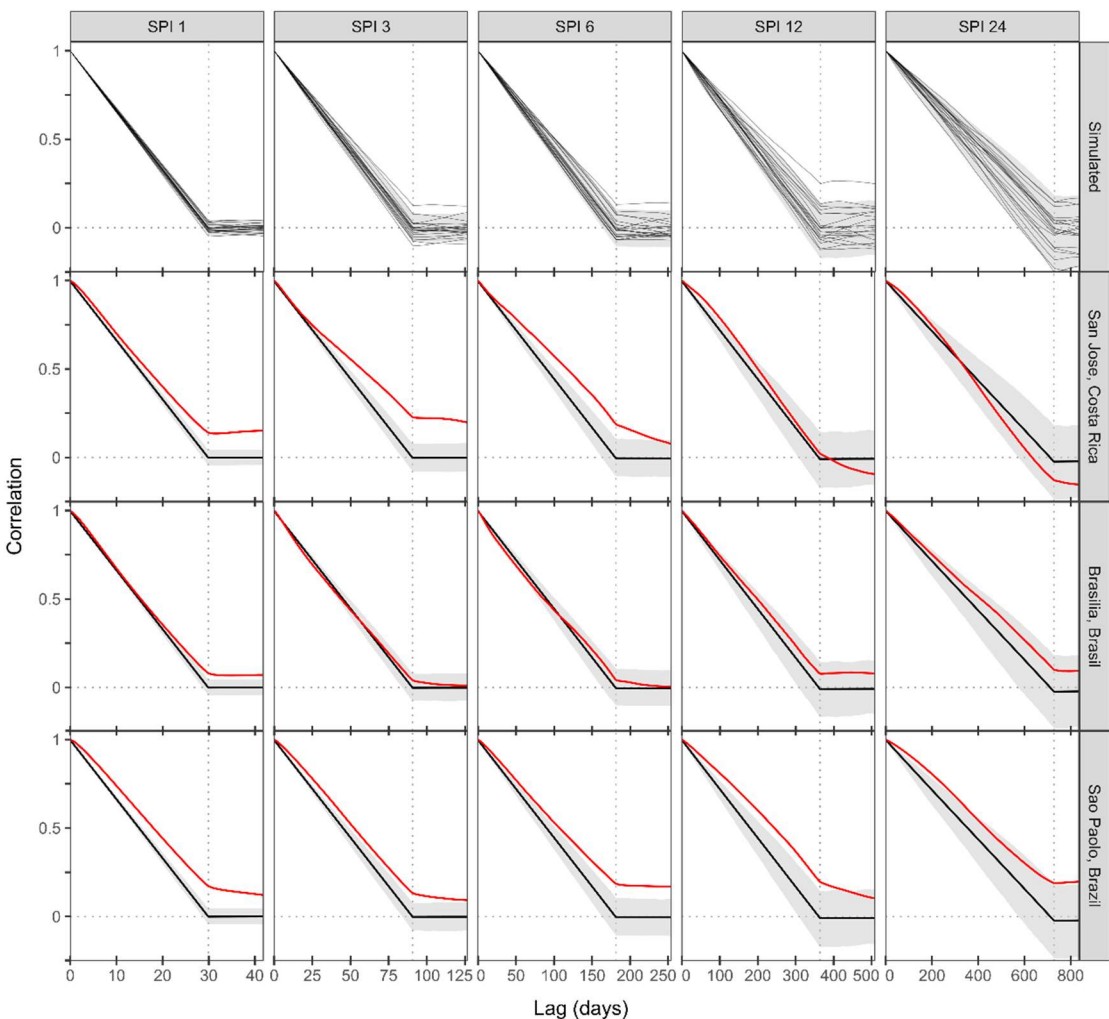

**Figure A10: Identical format to Fig 3 for representative grid cells in South America. Lagged correlation for the SPI-1, 3, 6, 12, and 24 moving windows. The first row shows 20 replicates from random simulation in light grey, while subsequent rows contrast observed autocorrelation (red) with theoretical (grey interval with black line).**



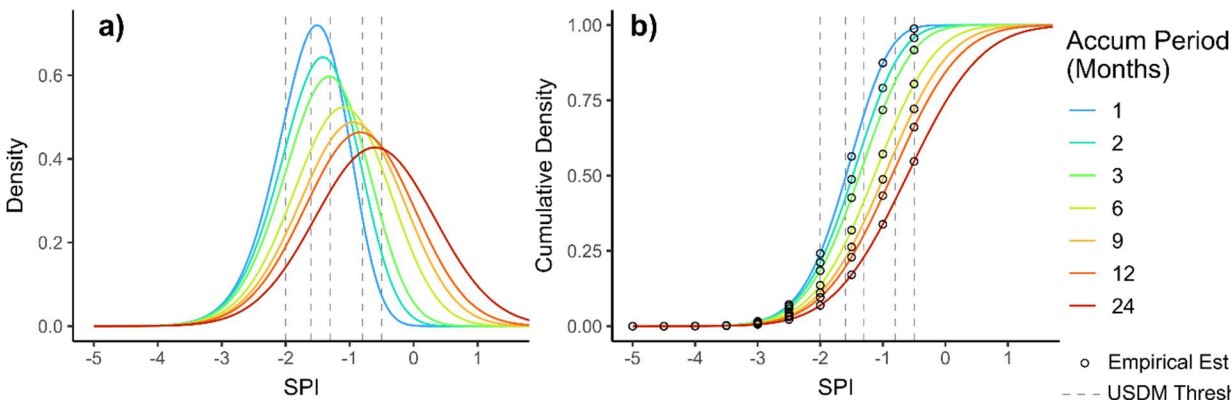

**Figure A11: Equivalent to Fig. 3, but calculated for monthly sequences. Annual minima (a) distribution and (b) cumulative probability density for monthly sequences of varied accumulation periods, indicated by colour. Colours are identical to Fig. 2. Vertical grey lines correspond to US Drought Monitor thresholds for D0-D4 (-0.5, -0.8, -1.3, -1.6, and -2.0). Open points represent empirical estimates directly from simulation.**


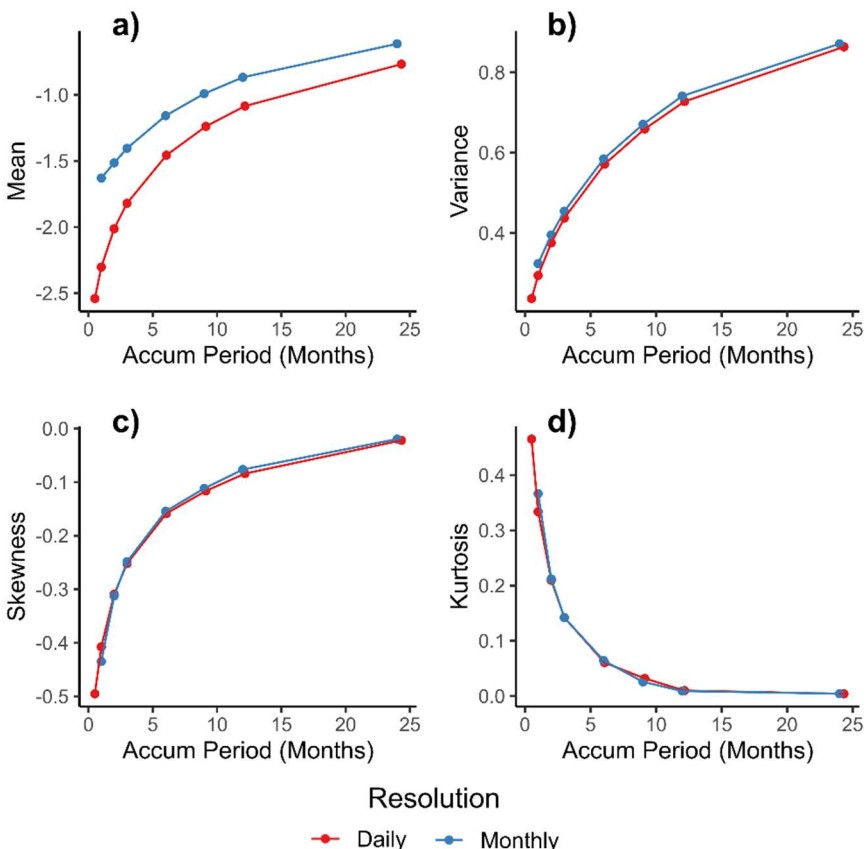


**Figure A12: The first four distribution moments for the annual minima from daily and monthly simulations (shown as colors). Accumulation period plotted on the x-axis.**


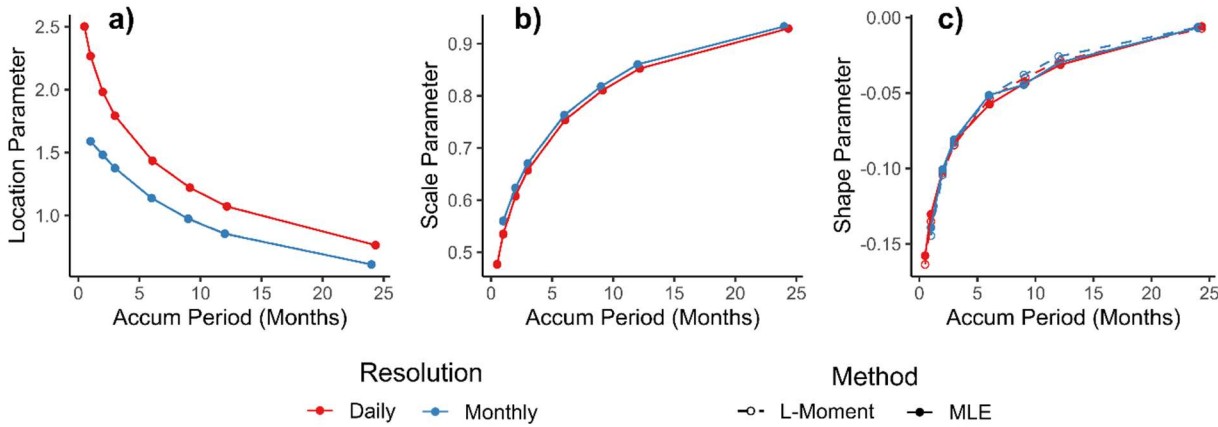

**Figure A13: Fitted distribution parameters for the Generalized Normal distribution describing annual minima from daily and monthly simulations (shown as colours). Accumulation period plotted on the x-axis. Fitting method are shown as linetype, though these are largely indistinguishable.**

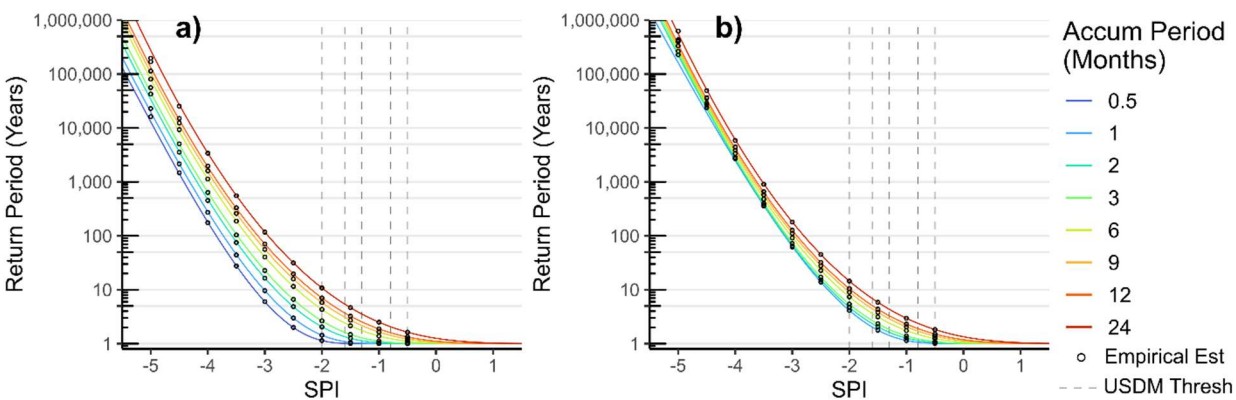

**Figure A14: Equivalent to Fig. 4, but expanded to SPI < -5. Return periods for (a) daily and (b) monthly sequences with accumulation periods indicated by colour. Colours are identical to Figs. 2 and 3. Vertical grey lines correspond to US Drought Monitor thresholds. Open points represent empirical estimates directly from simulation..**

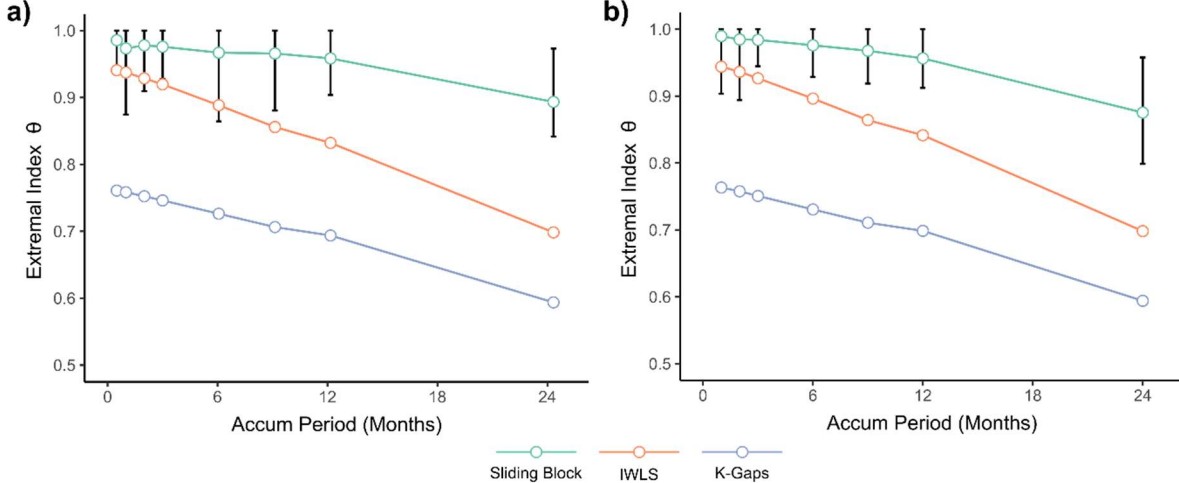


**Figure A15: Extremal index, θ, calculated for (a) daily and (b) monthly records. Colours represent three approaches for calculating the extremal index: sliding block maxima (Berghaus and Bucher 2018), iterated weight least squares (Suveges 2007), and the K-gaps model (Suvege and Davison 2010). Empty circles represent the mean of the extremal index for the twenty subsets, while bars represent the min and max extremal index values. Extremal index is bounded**

**between 0 and 1, with θ=1 representing completely independent extremes and values less than 1 suggesting increased extremal clustering. Appendix B. Additional Tables**

Table B1: Distribution parameters for the generalized normal distribution.

| Accumulation (months) | Daily | | | Monthly | | |
|---|---|---|---|---|---|---|
| | kappa | alpha | xi | kappa | alpha | xi |
| 0.5 | -0.1578 | 0.4773 | 2.503 | | | |
| 1 | -0.1303 | 0.5353 | 2.268 | -0.1391 | 0.5605 | 1.590 |
| 2 | -0.1015 | 0.6081 | 1.982 | -0.1006 | 0.6239 | 1.483 |
| 3 | -0.0820 | 0.6578 | 1.793 | -0.0808 | 0.6706 | 1.377 |
| 6 | -0.0574 | 0.7542 | 1.434 | -0.0515 | 0.7634 | 1.138 |
| 9 | -0.0432 | 0.8108 | 1.220 | -0.0445 | 0.8181 | 0.973 |
| 12 | -0.0313 | 0.8525 | 1.071 | -0.0299 | 0.8604 | 0.854 |
| 24 | -0.0059 | 0.9293 | 0.764 | -0.0068 | 0.9332 | 0.610 |

**Code Availability**

All data are available via an open-access repository (doi: 10.5281/zenodo.8331359). This code has been tested to generate all figures and tables presented in this paper.

**Data Availability**

All data are available via an open-access repository (doi: 10.5281/zenodo.8331359). This code has been tested to generate all figures and tables presented in this paper.

**Author Contribution**

JHS: Conceptualization, Methodology, Formal analysis, Writing – original draft
KS, IM, and MAIH: Conceptualization, Writing – review and editing.

**Competing Interests**

The authors declare that they have no conflict of interest.

**Disclaimer**

Any opinions, findings, and conclusions or recommendations expressed in this material are those of the author(s) and do not necessarily reflect those of the National Science Foundation.

**Acknowledgements**

The authors would like to thank Dr. Peter Craigmille for his discussion regarding this paper and some of its theoretical underpinnings.

This material is based upon work supported by the National Science Foundation under grant no. 2002539. The work was
also supported by the Byrd Polar and Climate Research Center at The Ohio State University.

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
