# Peer review of "Expected Annual Minima from an Idealized Moving Average Drought Index"

_EGUsphere, 2024_

## Author Response (AR1)

**Response to Reviewers**

*"Theoretical Annual Exceedances from Moving Average Drought Indices"*

**Reviewer #1:**

**General Comment**

The study presents a theoretical analysis of the annual return period of standardized variables (such as SPI) commonly used in drought study. I found the topic of the research of interest, as someone that fully support a better clarity on terms such as '100 years drought' often used in the community without robust statistical support. The paper is well structure and easy to follow. I have, however, two major concerns that make hard for me to recommend the publication of the paper in its current form:

> **Thank you. We share your concern about this terminology, which largely motivated the study. We have attempted to address your comments below and in the revised manuscript. All line numbers refer to the version with tracked changes.**

1) The authors generate 10 million years of data based on only two criteria: i) each month is standard normally distributed, and ii) a uniformly weighted backward average. As also stated by the authors, the only factor that generates autocorrelation in this procedure is the moving window, and any other factor causing persistence is ignored. This is a very strong assumption, as the scientific literature is full of studies on the tendency of rainfall to persist in the dry status, clustering of rainfall days, burstiness, etc. My question for the authors is: how much the theoretical time series generated with this approach resemble actual SPI time series? The authors do not provide any evidence that the theoretical values behave like real values, so any conclusion on the statistical behaviour of the theoretical data can be completely meaningless in real conditions, unless the author demonstrate that real and theoretical data are similar (statistically). My statistical background is not good enough to suggest a "validation" strategy (autocorrelograms?), but without this key step the results reported in this study cannot go beyond a mere mathematical exercise not suitable for a scientific publication.

> **Thank you for this comment. We had already begun a follow-on study to validate the temporal autocorrelation structure and identify where climatological persistence appears to exceed structural persistence. But, we were concerned that the number of permutations of indices (SPI, SPEI, SSI, SGI), moving window lengths, and locations around the world (gridded or stations) would make this analysis too extensive to include as a subsection within this study.**
>
> **We agree that most readers would like validation that this study has value for real-world SPI beyond simply moving average time series theory. We therefore have added a new analysis (Sections 2.3 and 3.2) that compares temporal autocorrelation at 23 representative sites across a range of climate zones with theoretical autocorrelation. We show that observed temporal autocorrelation typically falls**

**within the 95% interval for all sites, with only small deviations. This hopefully provides much-needed validation of our results for use in the real world. We still plan to perform a more spatially thorough analysis globally in a future study, and point out the need for this in the revised Discussion (Section 4.1) and the Conclusions (Lines 504-508).**

**Our goal for this study was to define a baseline derived from an idealized theoretical case in as general terms as possible, so that it could be applied to any moving average time series application. We found this solution lacking in the literature. As mentioned, we plan to build upon the theoretical case shown here and welcome others to do so with future studies about temporal autocorrelation and more drought-targeted metrics (see response to the next major comment).**

2) The way that SPI (or any other standardized index) is used in real applications is often for the detection of drought events (i.e., consecutive periods with values below a certain threshold). In this context, the analysis of annual minima is not really in line with what is commonly used by practitioners.

**We agree that more complex drought metrics could be tested following our approach. Similar to our response to the first comment, we have already begun a follow-up intended to test metrics like drought duration, frequency, and severity for multiple thresholds based on the time series model introduced here. However, we wanted to first establish the theory behind this model and to use the most general metric, block annual maxima/minima, that could be used outside of the specific SPI case.**

**We have added text (Lines 12, 95-100) and changed the title to highlight the study's focus on annual minima.**

To refer to one of the examples reported in the manuscript "The idea of experiencing an extreme flash drought at least once every other year…": no one is going to define as an "extreme drought event" a single isolated anomaly on a 15-day period (which, by the way, is a very unusual time window for anomaly computation).

**Our reference to flash drought was inexact and we have clarified our purpose for including the 15 day window (Lines 164-166) while also removing instances where we refer to this as a flash drought (Lines 84-85, 325, 331).**

**We agree that most flash drought definitions focus on the rate of change over a short period, rather than the absolute value during this short period. We do note that many definitions use 14 or 15 day windows (US Drought Monitor, Christian et al. 2019, Lisonbee et al. 2022), which agrees with our choice to consider a 15 day moving average. One could use the first derivative of the SPI, SPEI, or soil moisture over this 15 day period to define a flash drought, but we did do that here. So, we have softened our language in the paper, but retained the 15 day period as we still feel it has value in the context of extreme short duration drought metrics.**

The concept of consecutive time steps under a given threshold is a key factor in defining a drought, and, in this regard, your analysis on the annual minima may have very limited connection to what is commonly used from claims such as "a one in 100 years drought". A much more interesting analysis would be on the drought periods, and their effective (annual) return period (including the effects of inter-arrival time, etc). I understand that this may diverge too much from the goal of this study, but, at the minimum, the focus on annual minima (rather than event) should be super clear from the start (title, abstract, motivation, etc.) and the caveats in using these results when discussing events should be clarified.

**We strongly believe there is value in exploring the annual minima/return period, as it is the most commonly used extreme value metric generally, and because we find that authors still refer to return periods when discussing the SPI. Our hope is to better separate references to percentiles for individual months or days (the basis for the SPI) from return periods (Lines 492-496). As we show here, while the likelihood of an SPI less than -2 is unlikely, the likelihood of a single day in a year less than -2 is quite common.**

**As previously mentioned, we have added text (Lines 12, 95-100) and changed the title to highlight the study's focus on annual minima.**

I am aware that the authors have a clear view on the limitations of this study in regards of both topics, as evidenced in some parts of the discussion. However, I still believe that a proper analysis of the base assumptions of the study need to be added before considering valuable the obtained outcomes.

**We have added a new analysis of temporal autocorrelation to address your first major comment (Sections 2.3 and 3.2). Our intention with this study was to establish the theoretical underpinnings of the moving average model as a baseline reference. Now that we have established the underlying theory, we plan to expand to focus on real-world use cases with follow-up studies addressing: (1) global analysis of climatological vs structural persistence across multiple variables, and (2) more drought-specific metrics (Lines 503-510).**

Beyond these two major criticisms, I report some additional comments that I hope will be useful to improve the overall quality of the manuscript.

Title: the focus on annual minima should be clear already in the title.

**We agree and have modified the title to reflect this.**

L12. Same here, exceedance of annual minima…

**Agreed. We have changed this text.**

L17. Something on the extreme clustering should be mentioned here.
**We disagree that a discussion of extreme clustering should be included in the abstract. This is one hypothesis as to why we find a deviation from the GEV**

**distribution. We provide some initial evidence that this might be playing a role, but a more detailed statistical proof would be required to be certain. Because we cannot provide that here, we suggest to leave hypothesis out of the abstract and focus on our results.**

L112. There is nothing that backup this claim. As a key factor, a proof of this assumption is absolutely required.

> **We agree that this sentence was strong, without support. As stated for your first major comment, we have added new analysis (Sections 2.3 and 3.2 that supports the use of our theoretical model. We have modified Line 112 to soften this language and support it later.**

Fig. 1. Would an analogous plot for real SPI-6 values have a similar shape?

> **Similar to the previous response and Major comment 1, we have added Figures 3 and Supplemental Figures A6-A10 that demonstrate agreement with the theory for a multiple locations and window lengths globally. We intend to follow up this study with a complete global analysis.**

L175. "…is exactly symmetrical…" Is this statement true? It is true that SPI are derived from a standard normal distribution, but as a rescaling of a Gamma (usually, or any other left bounded distribution), SPI shouldn't be symmetrical. I do not think that this is a problem in your study, but I would be carefully rewording this sentence.

> **We have clarified that the underlying distribution may be skewed, but the resulting SPI distribution should be normal/symmetrical. Our statement that it is exactly symmetrical would be true if the Gamma distribution used to normalize the data produces a perfect fit. In practice there are inevitably deviations from thisso we have chosen to rewrite this sentence as "… is symmetrical" to acknowledge the potential for small deviations.**

L183. How did you define a good fitting based on the AIC? Which values? Significance?
> **AIC cannot be used to determine significance, but it can make a relative comparison. We have added text to Lines 263-264 and a figure (Fig. A4) confirming that Gen Normal has a consistently better fit than the GEV. This conclusion is even more strongly supported by the near identical moment matches and Q-Q plot.**

L191. This section is a little confusing. As currently stated, it may give the impression that a "normal" distribution is followed. However, in my understanding, the 3-parameter lognormal distribution is still a distribution designed to reproduce extremes, it is only not part of the GEV family. In the current form, it seems that the data do not follow the behaviour of extreme values but that of "normal" values, but this is not the case. I think the section need to be reworded to better clarify what does it mean (in practice) that the data follow the 3-parameter lognormal distribution rathe than the GEV. Most of the readers of the papers may not be expert in statistics and may come up with a wrong conclusion.

**We have modified this section (Lines 225-232) to better clarify the purpose of fitting the Generalized Normal and have tried to avoid talking about "normal" when we mean symmetrical/zero skew. Hopefully the additions here, combined with better clarification on the AIC, etc. have made this section more clear.**

L212. The log transformation is commonly deployed to use normal distribution on extreme values, so this is coherent with the extreme nature of annual minima.
**We agree with this statement.**

L215-216. It is not clear if these studies used the generalized normal for extreme values, similar to the ones analysed here.
**We have clarified that these studies used the generalized normal for extreme values.**

L232. It would be useful to report an example for the GEV too. Also, I don't see any AIC values reported. How did you use AIC to evaluate the goodness of fit? Are all the fittings statistically significant?.
**We have added text to Lines 273-275, confirming that AIC is lower for the Generalized Normal distribution than the GEV distribution for all accumulation periods. We refer to a new supplemental figure (Fig. A4) which shows this.**

**We purposefully avoided talking about statistical significance because it is not particularly meaningful when n = 10 million. Instead, we point to nearly identical higher order moments, an extremely accurate Q-Q plot, even for extremes in the million year return period, and consistently lower AIC values.**

L283. Stagge et al. (2016).

**We have corrected this.**

L288-295. Albeit true, this effect may be amplified by the particular method used to generate the data. In real SPI time series, some effect of persistency will be present even in SPI-1 or SPI-3, otherwise the concept of "drought event" would not be possible in such time series..

**We have added a sentence here (Lines 350-353) pointing readers to the new analysis of temporal autocorrelation in observed SPI time series (Section 3.2).**

**We agree that climate drivers and teleconnections can drive drought events, though there are also drought events driven by internal variability (randomness as simulated here).**

L307. For which SPI value? -2?
**Thank you, we have clarified this is for -2 (Lines 370-371) .**

L311-313. This is true only if annual minima are analysed. If events are analysed (consecutive periods under a certain threshold), then the daily or monthly time scale should only have minimal effects.

**We cannot be certain if this is the case because we have not tested drought events via a theory of runs style analysis. However, we intend to do this in a companion study currently being developed. We have added this statement to the Conclusions (Lines 504-510).**

L323. This should read "Discussion".

**Thank you. We have changed this.**

L344. Again, how can you confidently claim that those are minimal deviation, as they are an integral part of how precipitation behaves.

**We have added a new analysis (Section 3.2) to test the relative importance of structural vs climatological persistence and have added a new paragraph (Lines 408-415) clarifying these findings.**

L351. Berman (1964).

**This has been corrected.**

L365. A figure on the extreme clustering is needed to better show this concept and its effects. In general, this part seems really useful but very poorly presented and discussed.

**We have added a figure showing values of the extremal clustering index, θ (Fig. A15). This confirms minor levels of extremal clustering (in absolute terms), but the pattern of increased clustering mirrors the pattern of increased deviation from the GEV distribution (Lines 437-440).**

**We have also modified the text to make it clear that this extremal clustering explanation is one hypothesis (Lines 427-428) and would require new analyses or statistical derivations to confirm this hypothesis (Lines 440-441). Such an analysis is beyond the scope of this paper, but we strongly hope that future research will explore the underlying causal mechanism.**

L371-372. This is true for long accumulation periods only.

**Agreed. We have clarified this in the edited text.**

L375. Again, a reference to extreme clustering is made but without support of data showing the effect to the readers.

**We have added an extremal clustering index figure (Fig. A15) to support this claim.**

L379. There is a typo.

**We checked this line and those surrounding it but could not find a typo.**

L383. Again, the focus on annual minima is not clear here. Annual exceedance probability can be computed for any metrics derived from SPI time series.

**We have modified this sentence (Lines 458-459) to reflect an idealized time series and annual minima.**

L400. It should be observatory (also in other sections of the text).

**Thank you. We have a done a search and corrected all instances.**

L418. …period).

**Thank you. This has been corrected.**

L419-420. This is related to a common understanding of how probability works, and it has very little to do with the results of your study. Similarly, someone could argue that saying that a minimum annual SPI < -2 has a certain occurrence probability (as reported in this study) is completely different than the probability of a certain drought event (with a given severity) to occur.

**We do not agree that this is immediately clear to all readers and believe there is value in clarifying it. Our concern is that readers might mix the two concepts, probability of exceeding -2 on a given day vs probability of exceeding -2 in a given year. We have tried to clarify our point in the revised document (Lines 493-497).**

**We also added a final recommendation that future studies should explore more specific drought event statistics (Lines 504-510).**

**Reviewer #2 comments:**

The manuscript „Theoretical Annual Exceedances from Moving Average Drought Indices" applies a statistical model for approximating return periods for moving averages processes of standardized drought indices. The study highlights the problem of the terminology of return periods in case of drought indices and provides theoretical values for these return periods by a simulation study. I think the study provides a novel contribution scientific literature, especially for the application (and misuse) of standardized drought indices. Nevertheless, I have some major concerns, which hopefully can be addressed by the authors, as they make an important point about the use of drought indices, which is worth publishing.

**Thank you for your support and your review. We have addressed your comments below and in the revised document. All line numbers refer to the version with tracked changes.**

First, I find it problematic, that the temporal autocorrelation of the drought index is outlined, but then a distribution is fitted to the annual minima series, where the temporal correlation is not considered prior to fitting the distribution. In my opinion the study would benefit, if a statistical sound estimation procedure is outlined, which considers the temporal autocorrelation before fitting an extreme value distribution. This does not mean, to actually change the entire scope of the study, but to provide a better approach for many applications.

**We have added an entirely new analysis, Sections 2.3 (Methods) and 3.2 (Results) to confirm the temporal autocorrelation structure prior to fitting the distribution for annual minima.**

Second, I think the Introduction would benefit of highlighting, where in scientific literature this pitfall of annual return periods for drought indices, actually can be identified. As drought events are mainly characterized over intensity, duration, I think it is important to underline, where return periods of annual SPI values are estimated to better frame the scope of the study.

**We purposefully avoided naming specific researchers or studies in the Introduction. Rather, we chose to explain the correct way to describe return periods.**

**Our literature survey did identify studies that compare return periods across regions, but far more that compare drought duration and other Theory of Run drought metrics across regions, that theoretically should not vary dramatically. These could also be calculated following the stochastic approach outlined here and we have already begun this research as a subsequent study. Because it would be too much to include here, we have instead only included a recommendation for such a study in the Conclusions (Line 498-505). For this first study, we have tried to keep our analysis as general as possible, focusing only on annual minima/maxima. We believe this would make our study useful for any Moving Average time series analysis.**

Finally, the simulation study could be set up in a more realistic scenario, which includes sample sizes that are currently common for e.g. precipitation, streamflow or groundwater time series. In this context, also the uncertainty of these moving averages can be quantified. I think it is not the best solution to provide values for these return periods, but to outline a more robust statistical estimation procedure.

**Our goal with this study was to determine the true expected value from this idealized time series, hence the millions of simulated years. You make a good point that for much shorter time series, the confidence intervals would increase, but still be centered around these expected values. Such an analysis would be similar to how we developed confidence intervals for the temporal autocorrelation in Section 3.2 and Figs. 3, A6-10.**

**Rather than performing this analysis for a fixed time series length, because any dataset would have a different length, we hope that authors can use our stochastic model (Eqs. 2-3) to simulate confidence intervals for their specific use case.**

Minor comments:

Line 63: Maybe this sentence could be split up.

**Thank you, we have split this sentence (Line 64-65).**

Line 181: Which method was then actually used (MLE or l-moments)? I believe there should be no difference in the estimates.

**Thank you, we have clarified that we used l-moments (Lines 213-214).**

Line 185: Please specify the actual used exceedance probabilites somewhere in the methods section.

**We have clarified that all later estimates are using l-moments (Line 216).**

Line 220: Rootzen, 1986).

**Thank you. We have corrected this.**

Line 305: Is this not only an effect of sample size (and the temporal autocorrelation)? As I have 365 chances of the SPI value falling below -2 (daily data), or 12 chances of a fixed window falling below -2 (monthly data)?

**We agree and this was partially our point, but we did not state it explicitly/clearly in the text. We have added a sentence to state this (Lines 370-371).**

Line 349: Missing bracket.

**Thank you. We have corrected this.**

---

## Author Response (AR2)

**Response to Reviewers**

*"Expected Annual Minima from an Idealized Moving Average Drought Index"*

**Editor:**

Thank you for handling our manuscript. We have done an additional round of proofreading and addressed the specific grammar error you pointed out.